# BUILDING NORMALIZING FLOWS WITH STOCHASTIC INTERPOLANTS

**Michael S. Albergo**
Center for Cosmology and Particle Physics
New York University
New York, NY 10003, USA
albergo@nyu.edu

**Eric Vanden-Eijnden**
Courant Institute of Mathematical Sciences
New York University
New York, NY 10012, USA
eve2@cims.nyu.edu

## ABSTRACT

A generative model based on a continuous-time normalizing flow between any pair of base and target probability densities is proposed. The velocity field of this flow is inferred from the probability current of a time-dependent density that interpolates between the base and the target in finite time. Unlike conventional normalizing flow inference methods based the maximum likelihood principle, which require costly backpropagation through ODE solvers, our interpolant approach leads to a simple quadratic loss for the velocity itself which is expressed in terms of expectations that are readily amenable to empirical estimation. The flow can be used to generate samples from either the base or target, and to estimate the likelihood at any time along the interpolant. In addition, the flow can be optimized to minimize the path length of the interpolant density, thereby paving the way for building optimal transport maps. In situations where the base is a Gaussian density, we also show that the velocity of our normalizing flow can also be used to construct a diffusion model to sample the target as well as estimate its score. However, our approach shows that we can bypass this diffusion completely and work at the level of the probability flow with greater simplicity, opening an avenue for methods based solely on ordinary differential equations as an alternative to those based on stochastic differential equations. Benchmarking on density estimation tasks illustrates that the learned flow can match and surpass conventional continuous flows at a fraction of the cost, and compares well with diffusions on image generation on CIFAR-10 and ImageNet $32 \times 32$. The method scales ab-initio ODE flows to previously unreachable image resolutions, demonstrated up to $128 \times 128$.

## 1 INTRODUCTION

Contemporary generative models have primarily been designed around the construction of a map between two probability distributions that transform samples from the first into samples from the second. While progress has been from various angles with tools such as implicit maps (Goodfellow et al., 2014; Brock et al., 2019), and autoregressive maps (Menick & Kalchbrenner, 2019; Razavi et al., 2019; Lee et al., 2022), we focus on the case where the map has a clear associated *probability flow*. Advances in this domain, namely from flow and diffusion models, have arisen through the introduction of algorithms or inductive biases that make learning this map, and the Jacobian of the associated change of variables, more tractable. The challenge is to choose what structure to impose on the transport to best reach a complex target distribution from a simple one used as base, while maintaining computational efficiency.

In the continuous time perspective, this problem can be framed as the design of a time-dependent map, $X_t(x)$ with $t \in [0, 1]$, which functions as the push-forward of the base distribution at time $t = 0$ onto some time-dependent distribution that reaches the target at time $t = 1$. Assuming that these distributions have densities supported on $\Omega \subseteq \mathbb{R}^d$, say $\rho_0$ for the base and $\rho_1$ for the target, this amounts to constructing $X_t : \Omega \to \Omega$ such that

$$\text{if } x \sim \rho_0 \text{ then } X_t(x) \sim \rho_t \text{ for some density } \rho_t \text{ such that } \rho_{t=0} = \rho_0 \text{ and } \rho_{t=1} = \rho_1. \quad (1)$$

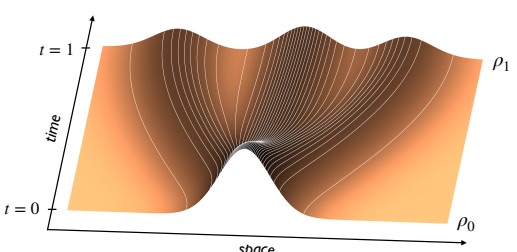

Figure 1: The density $\rho_t(x)$ produced by the stochastic interpolant based on (5) between a standard Gaussian density and a Gaussian mixture density with three modes. Also shows in white are the flow lines of the map $X_t(x)$ our method produces.

One convenient way to represent this time-continuous map is to define it as the flow associated with the ordinary differential equation (ODE)

$$\dot{X}_t(x) = v_t(X_t(x)), \qquad X_{t=0}(x) = x \qquad (2)$$

where the dot denotes derivative with respect to $t$ and $v_t(x)$ is the *velocity field* governing the transport. This is equivalent to saying that the probability density function $\rho_t(x)$ defined as the push-forward of the base $\rho_0(x)$ by the map $X_t$ satisfies the continuity equation (see e.g. (Villani, 2009; Santambrogio, 2015) and Appendix A)

$$\partial_t \rho_t + \nabla \cdot (v_t \rho_t) = 0 \qquad \text{with} \ \ \rho_{t=0} = \rho_0 \ \ \text{and} \ \ \rho_{t=1} = \rho_1, \qquad (3)$$

and the inference problem becomes to estimate a velocity field such that (3) holds.

Here we propose a solution to this problem based on introducing a time-differentiable interpolant

$$I_t : \Omega \times \Omega \rightarrow \Omega \ \ \text{such that} \ \ I_{t=0}(x_0, x_1) = x_0 \ \ \text{and} \ \ I_{t=1}(x_0, x_1) = x_1 \qquad (4)$$

A useful instance of such an interpolant that we will employ is

$$I_t(x_0, x_1) = \cos(\tfrac{1}{2}\pi t)x_0 + \sin(\tfrac{1}{2}\pi t)x_1, \qquad (5)$$

though we stress the framework we propose applies to any $I_t(x_0, x_1)$ satisfying (4) under mild additional assumptions on $\rho_0$, $\rho_1$, and $I_t$ specified below. Given this interpolant, we then construct the stochastic process $x_t$ by sampling independently $x_0$ from $\rho_0$ and $x_1$ from $\rho_1$, and passing them through $I_t$:

$$x_t = I_t(x_0, x_1), \qquad x_0 \sim \rho_0, \quad x_1 \sim \rho_1 \quad \text{independent.} \qquad (6)$$

We refer to the process $x_t$ as a *stochastic interpolant*. Under this paradigm, we make the following key observations as our **main contributions** in this work:

- The probability density $\rho_t(x)$ of $x_t$ connecting the two densities, henceforth referred to as the *interpolant density*, satisfies (3) with a velocity $v_t(x)$ which is the unique minimizer of a simple quadratic objective. This result is the content of Proposition 1 below, and it can be leveraged to estimate $v_t(x)$ in a parametric class (e.g. using deep neural networks) to construct a generative model through the solution of the probability flow equation (2), which we call InterFlow.

- By specifying an interpolant density, the method therefore separates the tasks of minimizing the objective from discovering a path between the base and target densities. This is in contrast with conventional maximum likelihood (MLE) training of flows where one is forced to couple the choice of path in the space of measures to maximizing the objective.

- We show that the Wasserstein-2 ($W_2$) distance between the target density $\rho_1$ and the density $\hat{\rho}_1$ obtained by transporting $\rho_0$ using an approximate velocity $\hat{v}_t$ in (2) is controlled by our objective function. We also show that the value of the objective on $\hat{v}_t$ during training can be used to check convergence of this learned velocity field towards the exact $v_t$.

- We show that our approach can be generalized to shorten the path length of the interpolant density and optimize the transport by additionally maximizing our objective over the interpolant $I_t(x_0, x_1)$ and/or adjustable parameters in the base density $\rho_0$.

- By choosing $\rho_0$ to be a Gaussian density and using (5) as interpolant, we show that the score of the interpolant density, $\nabla \log \rho_t$, can be explicitly related to the velocity field $v_t$. This allows us to draw connection between our approach and score-based diffusion models, providing theoretical groundwork for future exploration of this duality.

- We demonstrate the feasibility of the method on toy and high dimensional tabular datasets, and show that the method matches or supersedes conventional ODE flows at lower cost, as it avoids the need to backpropagate through ODE solves. We demonstrate our approach on image generation for CIFAR-10 and ImageNet 32x32 and show that it scales well to larger sizes, e.g. on the 128×128 Oxford flower dataset.

| Methods | Map Type | Finite Time Integration | Simulation-Free Training | Computable Likelihood | Optimizable Transport |
|---|---|---|---|---|---|
| FFJORD | D | ✓ | $x$ | ✓ | ✓ |
| ScoreFlows | S/D | $x$ | ✓ | ✓ | $x$ |
| Schrödinger Bridge | S | ✓ | $x$ | $x$ | ✓ |
| InterFlow (Ours) | D | ✓ | ✓ | ✓ | ✓ |

Table 1: Description of the qualities of continuous time transport methods defined by a stochastic or deterministic process.

## 1.1 RELATED WORKS

Early works on exploiting transport maps for generative modeling go back at least to Chen & Gopinath (2000), which focuses on normalizing a dataset to infer its likelihood. This idea was brought closer to contemporary use cases through the work of Tabak & Vanden-Eijnden (2010) and Tabak & Turner (2013), which devised to expressly map between densities using simple transport functions inferred through maximum likelihood estimation (MLE). These transformations were learned in sequence via a greedy procedure. We detail below how this paradigm has evolved in the case where the map is represented by a neural network and optimized accordingly.

**Discrete and continuous time flows.** The first success of normalizing flows with neural network parametrizations follow the work of Tabak & Turner (2013) with a finite set of steps along the map. By imposing structure on the transformation so that it remains an efficiently invertible diffeomorphism, the models of Rezende & Mohamed (2015); Dinh et al. (2017); Huang et al. (2018); Durkan et al. (2019) can be optimized through maximum likelihood estimation at the cost of limiting the expressive power of the representation, as the Jacobian of the map must be kept simple to calculate the likelihood. Extending this to the continuous case allowed the Jacobian to be unstructured yet still estimable through trace estimation techniques (Chen et al., 2018; Grathwohl et al., 2019; Hutchinson, 1989). Yet, learning this map through MLE requires costly backpropagation through numerical integration. Regulating the path can reduce the number of solver calls (Finlay et al., 2020; Onken et al., 2021), though this does not alleviate the main structural challenge of the optimization. Our work uses a continuous map $X_t$ as well but allows for direct estimation of the underlying velocity. While recent work has also considered simulation-free training by fitting a velocity field, these works present scalability issues (Rozen et al., 2021) and biased optimization (Ben-Hamu et al., 2022), and are limited to manifolds. Moreover, (Rozen et al., 2021) relies on interpolating directly the probability measures, which can lead to unstable velocities.

**Score-based flows.** Adjacent research has made use of diffusion processes, commonly the Ornstein-Uhlenbeck (OU) process, to connect the target $\rho_1$ to the base $\rho_0$. In this case the transport is governed by a stochastic differential equation (SDE) that is evolved for infinite time, and the challenge of learning a generative model can be framed as fitting the reverse time evolution of the SDE from Gaussian noise back to $\rho_1$ (Sohl-Dickstein et al., 2015; Ho et al., 2020; Song et al., 2021b). Doing so indirectly learns the velocity field by means of learning the *score function* $\nabla \log \rho_t(x)$, using the Fischer divergence instead of the MLE objective. While this approach has shown great promise to model high dimensional distributions (Rombach et al., 2022; Hoogeboom et al., 2022), particularly in the case of text-to-image generation (Ramesh et al., 2022; Saharia et al., 2022), there is an absence of theoretical motivation for the SDE-flow framework and the complexity it induces. Namely, the SDE must evolve for infinite time to connect the distributions, the parameterization of the time steps remains heuristic (Xiao et al., 2022), and the criticality of noise, as well as the score, is not absolutely apparent (Bansal et al., 2022; Lu et al., 2022). In particular, while the objective used in score-based diffusion models was shown to bound the Kullback-Leibler divergence (Song et al., 2021a), actual calculation of the likelihood requires one to work with the ODE probability flow associated with the SDE. This motivates further research into effective, ODE-driven, approaches to learning the map. Our approach can be viewed as an alternative to score-based diffusion models in which the ODE velocity is learned through the interpolant $x_t$ rather than an OU process, leading to greater simplicity and flexibility (as we can connect any two densities exactly over a finite time interval).

**Bridge-based methods.** Heng et al. (2021) propose to learn Schroedinger bridges, which are a entropic regularized version of the optimal transportation plan connecting two densities in finite time, using the framework of score-based diffusion. Similarly, Peluchetti (2022) investigates the use of bridge processes, i.e. SDE whose position is constrained both at the initial and final times, to perform exact density interpolation in finite time. A key difference between these approaches and ours is that they give diffusion-based models, whereas our method builds a probability flow ODE directly using a quadratic loss for its velocity, which is simpler and shown here to be scalable.

**Interpolants.** Co-incident works by Liu et al. (2022); Lipman et al. (2022) derive an analogous optimization to us, with a focus on straight interpolants, also contrasting it with score-based methods. Liu et al. (2022) describe an iterative way of rectifying the interpolant path, which can be shown to arrive at an optimal transport map when the procedure is repeated *ad infinitum* (Liu, 2022). We also

propose a solution to the problem of optimal transport that involves optimizing our objective over the stochastic interpolant.

## 1.2 NOTATIONS AND ASSUMPTIONS

We assume that the base and the target distribution are both absolutely continuous with respect to the Lebesgue measure on $\mathbb{R}^d$, with densities $\rho_0$ and $\rho_1$, respectively. We do not require these densities to be positive everywhere on $\mathbb{R}^d$, but we assume that $\rho_0(x)$ and $\rho_1(x)$ are continuously differentiable in $x$. Regarding the interpolant $I_t : \mathbb{R}^d \times \mathbb{R}^d \to \mathbb{R}^d$, we assume that it is surjective for all $t \in [0, 1]$ and satisfies (4). We also assume that $I_t(x_0, x_1)$ is continuously differentiable in $(t, x_0, x_1)$, and that it is such that

$$\mathbb{E}\big[|\partial_t I_t(x_0, x_1)|^2\big] < \infty \tag{7}$$

A few additional technical assumptions on $\rho_0$, $\rho_1$ and $I_t$ are listed in Appendix B. Given any function $f_t(x_0, x_1)$ we denote

$$\mathbb{E}[f_t(x_0, x_1)] = \int_0^1 \int_{\mathbb{R}^d \times \mathbb{R}^d} f_t(x_0, x_1)\rho_0(x_0)\rho_1(x_1)dx_0 dx_1 dt \tag{8}$$

its expectation over $t$, $x_0$, and $x_1$ drawn independently from the uniform density on $[0, 1]$, $\rho_0$, and $\rho_1$, respectively. We use $\nabla$ to denote the gradient operator.

## 2 STOCHASTIC INTERPOLANTS AND ASSOCIATED FLOWS

Our first main theoretical result can be phrased as follows:

**Proposition 1.** *The stochastic interpolant $x_t$ defined in (6) with $I_t(x_0, x_1)$ satisfying (4) has a probability density $\rho_t(x)$ that satisfies the continuity equation (3) with a velocity $v_t(x)$ which is the unique minimizer over $\hat{v}_t(x)$ of the objective*

$$G(\hat{v}) = \mathbb{E}\left[|\hat{v}_t(I_t(x_0, x_1))|^2 - 2\partial_t I_t(x_0, x_1) \cdot \hat{v}_t(I_t(x_0, x_1))\right] \tag{9}$$

*In addition the minimum value of this objective is given by*

$$G(v) = -\mathbb{E}\big[|v_t(I_t(x_0, x_1))|^2\big] = -\int_0^1 \int_{\mathbb{R}^d} |v_t(x)|^2 \rho_t(x)dxdt > -\infty \tag{10}$$

Proposition 1 is proven in Appendix B under Assumption B.1. As this proof shows, the first statement of the proposition remains true if the expectation over $t$ is performed using any probability density $\omega(t) > 0$, which may prove useful in practice. We now describe some primary facts resulting from this proposition, itemized for clarity:

- The objective $G(\hat{v})$ is given in terms of an expectation that is amenable to empirical estimation given samples $t$, $x_0$, and $x_1$ drawn from $\rho_0$, $\rho_1$ and $U([0, 1])$. Below, we will exploit this property to propose a numerical scheme to perform the minimization of $G(\hat{v})$.

- While the minimizer of the objective $G(\hat{v})$ is not available analytically in general, a notable exception is when $\rho_0$ and $\rho_1$ are Gaussian mixture densities and we use the trigonometric interpolant (5) or generalization thereof, as discussed in Appendix C.

- The minimal value in (10) achieved by the objective implies that a necessary (albeit not sufficient) condition for $\hat{v} = v$ is

$$\tilde{G}(\hat{v}) = G(\hat{v}) + \mathbb{E}\big[|\hat{v}_t(I_t(x_0, x_1))|^2\big] = 0. \tag{11}$$

In our numerical experiments we will monitor this quantity. This minimal value also suggests to maximize $G(v) = \min_{\hat{v}} G(\hat{v})$ with respect to additional control parameters (e.g. the interpolant) to shorten the $W_2$ length of the path $\{\rho_t(x) : t \in [0, 1]\}$. In Appendix D, we show that this procedure achieves optimal transport under minimal assumptions.

- The last bound in (10), which is proven in in Lemma B.2, implies that the path length is always finite, even if it is not the shortest possible.

Let us now provide some intuitive derivation of the statements of Proposition 1:

**Continuity equation.** By definition of the stochastic interpolant $x_t$ we can express its density $\rho_t(x)$ using the Dirac delta distribution as

$$\rho_t(x) = \int_{\mathbb{R}^d \times \mathbb{R}^d} \delta\left(x - I_t(x_0, x_1)\right) \rho_0(x_0)\rho_1(x_1)dx_0dx_1. \tag{12}$$

Since $I_{t=0}(x_0, x_1) = x_0$ and $I_{t=1}(x_0, x_1) = x_1$ by definition, we have $\rho_{t=0} = \rho_0$ and $\rho_{t=1} = \rho_1$, which means that $\rho_t(x)$ satisfies the boundary conditions at $t = 0, 1$ in (3). Differentiating (12) in time using the chain rule gives

$$\partial_t\rho_t(x) = -\int_{\mathbb{R}^d \times \mathbb{R}^d} \partial_t I_t(x_0, x_1) \cdot \nabla\delta\left(x - I_t(x_0, x_1)\right) \rho_0(x_0)\rho_1(x_1)dx_0dx_1 \equiv -\nabla \cdot j_t(x) \tag{13}$$

where we defined the probability current

$$j_t(x) = \int_{\mathbb{R}^d \times \mathbb{R}^d} \partial_t I_t(x_0, x_1)\delta\left(x - I_t(x_0, x_1)\right) \rho_0(x_0)\rho_1(x_1)dx_0dx_1. \tag{14}$$

Therefore if we introduce the velocity $v_t(x)$ via

$$v_t(x) = \begin{cases} j_t(x)/\rho_t(x) & \text{if } \rho_t(x) > 0, \\ 0 & \text{else} \end{cases} \tag{15}$$

we see that we can write (13) as the continuity equation in (3).

**Variational formulation.** Using the expressions in (12) and (14) for $\rho_t(x)$ and $j_t(x)$ shows that we can write the objective (9) as

$$G(\hat{v}) = \int_0^1 \int_{\mathbb{R}^d} \left(|\hat{v}_t(x)|^2 \rho_t(x) - 2\hat{v}_t(x) \cdot j_t(x)\right) dxdt \tag{16}$$

Since $\rho_t(x)$ and $j_t(x)$ have the same support, the minimizer of this quadratic objective is unique for all $(t, x)$ where $\rho_t(x) > 0$ and given by (15).

**Minimum value of the objective.** Let $v_t(x)$ be given by (15) and consider the alternative objective

$$\begin{aligned} H(\hat{v}) &= \int_0^1 \int_{\mathbb{R}^d} |\hat{v}_t(x) - v_t(x)|^2 \rho_t(x)dxdt \\ &= \int_0^1 \int_{\mathbb{R}^d} \left(|\hat{v}_t(x)|^2\rho_t(x) - 2\hat{v}_t(x) \cdot j_t(x) + |v_t(x)|^2\rho_t(x)\right) dxdt \end{aligned} \tag{17}$$

where we expanded the square and used the identity $v_t(x)\rho_t(x) = j_t(x)$ to get the second equality. The objective $G(\hat{v})$ given in (16) can be written in term of $H(\hat{v})$ as

$$G(\hat{v}) = H(\hat{v}) - \int_0^1 \int_{\mathbb{R}^d} |v_t(x)|^2\rho_t(x)dxdt = H(\hat{v}) - \mathbb{E}\left[|v_t(I_t(x_0, x_1))|^2\right] \tag{18}$$

The equality in (10) follows by evaluating (18) at $\hat{v}_t(x) = v_t(x)$ using $H(v) = 0$.

**Optimality gap.** The argument above shows that we can use $\mathbb{E}\left[|\hat{v}_t(I_t(x_0, x_1)) - \partial_t I_t(x_0, x_1)|^2\right]$ as alternative objective to $G(\hat{v})$ since their first variations coincide. However, it should be stressed that this quadratic objective remains strictly positive at $\hat{v} = v$ in general so it offers no baseline measure of convergence. To see why, complete the square in $G(\hat{v})$ to write (18) as

$$H(\hat{v}) = \mathbb{E}\left[|\hat{v}_t(I_t) - \partial_t I_t|^2\right] - \mathbb{E}\left[|\partial_t I_t|^2\right] + \mathbb{E}\left[|v_t(I_t)|^2\right] \geq -\mathbb{E}\left[|\partial_t I_t|^2\right] + \mathbb{E}\left[|v_t(I_t)|^2\right] \tag{19}$$

where we used the shorthand notation $I_t = I_t(x_0, x_1)$ and $\partial_t I_t = \partial_t I_t(x_0, x_1)$. Evaluating this inequality at $\hat{v} = v$ using $H(v) = 0$ we deduce

$$\mathbb{E}\left[|\partial_t I_t|^2\right] \geq \mathbb{E}\left[|v_t(I_t)|^2\right] \tag{20}$$

However we stress that this inequality is not saturated, i.e. $\mathbb{E}\left[|v_t(I_t)|^2\right] \neq \mathbb{E}\left[|\partial_t I_t|^2\right]$, in general (see Remark B.3). Hence

$$\begin{aligned} \mathbb{E}\left[|v_t(I_t) - \partial_t I_t|^2\right] &= \min_{\hat{v}} \mathbb{E}\left[|\hat{v}_t(I_t) - \partial_t I_t|^2\right] \\ &= \min_{\hat{v}} G(\hat{v}) + \mathbb{E}\left[|\partial_t I_t|^2\right] = -\mathbb{E}\left[|v_t(I_t)|^2\right] + \mathbb{E}\left[|\partial_t I_t|^2\right] \geq 0. \end{aligned} \tag{21}$$

**Optimizing the transport.** It is natural to ask whether our stochastic interpolant construction can be amended or generalized to derive optimal maps. Here, we state a positive answer to this question by showing that the maximizing the objective $G(\hat{v})$ in (9) with respect to the interpolant yields a solution to the optimal transport problem in the framework of Benamou & Brenier (2000). This is proven in Appendix D, where we also discuss how to shorten the path length in density space by optimizing adjustable parameters in the base density $\rho_0$. Experiments are given in Appendix H. Since our primary aim here is to construct a map $T = X_{t=1}$ that pushes forward $\rho_0$ onto $\rho_1$, but not necessarily to identify the optimal one, we leave the full investigation of the consequences of these results for future work, but state the proposition explicitly here. In their seminal paper, Benamou & Brenier (2000) showed that finding the optimal map requires solving the minimization problem

$$\min_{(\hat{v},\hat{\rho})} \int_0^1 \int_{\mathbb{R}^d} |\hat{v}_t(x)|^2 \hat{\rho}_t(x) dx dt \tag{22}$$

$$\text{subject to:} \quad \partial_t \hat{\rho}_t + \nabla \cdot (\hat{v}_t \hat{\rho}_t) = 0, \quad \hat{\rho}_{t=0} = \rho_0, \quad \hat{\rho}_{t=1} = \rho_1.$$

The minimizing coupling $(\rho_t^*, \phi_t^*)$ for gradient field $v_t^*(x) = \nabla \phi_t^*(x)$ is unique and satisfies:

$$\partial_t \rho_t^* + \nabla \cdot (\nabla \phi_t^* \rho_t^*) = 0, \quad \rho_{t=0}^* = \rho_0, \quad \rho_{t=1}^* = \rho_1, \quad \partial_t \phi_t^* + \tfrac{1}{2}|\nabla \phi_t^*|^2 = 0. \tag{23}$$

In the interpolant flow picture, $\rho_t(x)$ is fixed by the choice of interpolant $I_t(x_0, x_1)$, and in general $\rho_t(x) \neq \rho_t^*(x)$. Because the value of the objective in (22) is equal to the minimum of $G(\hat{v})$ given in (10), a natural suggestion to optimize the transport is to maximize this minimum over the interpolant. Under some assumption on the Benamou-Brenier density $\rho_t^*(x)$ solution of (23), this procedure works. We show this through the use of *interpolable densities* as discussed in Mikulincer & Shenfeld (2022) and defined in D.1.

**Proposition 2.** *Assume that (i) the optimal density function $\rho_t^*(x)$ minimizing (22) is interpolable and (ii) (23) has a classical solution. Consider the max-min problem*

$$\max_{\hat{I}} \min_{\hat{v}} G(\hat{v}) \tag{24}$$

*where $G(\hat{v})$ is the objective in (9) and the maximum is taken over interpolants satisfying (4). Then a maximizer of (24) exists, and any maximizer $I_t^*(x_0, x_1)$ is such that the probability density function of $x_t^* = I_t^*(x_0, x_1)$, with $x_0 \sim \rho_0$ and $x_1 \sim \rho_1$ independent, is the optimal $\rho_t^*(x)$, the mimimizing velocity is $v_t^*(x) = \nabla \phi_t^*(x)$, and the pair $(\rho_t^*(x), \phi_t^*(x))$ satisfies (23).*

The proof of Proposition 2 is given in Appendix D, along with further discussion. Proposition 2 relies on Lemma D.3 that reformulates (22) in a way which shows this problem is equivalent to the max-min problem in (24) for interpolable densities.

## 2.1 WASSERSTEIN BOUNDS

The following result shows that the objective in (17) controls the Wasserstein distance between the target density $\rho_1$ and the the density $\hat{\rho}_1$ obtained as the pushforward of the base density $\rho_0$ by the map $\hat{X}_{t=1}$ associated with the velocity $\hat{v}_t$:

**Proposition 3.** *Let $\rho_t(x)$ be the exact interpolant density defined in (12) and, given a velocity field $\hat{v}_t(x)$, let us define $\hat{\rho}_t(x)$ as the solution of the initial value problem*

$$\partial_t \hat{\rho}_t + \nabla \cdot (\hat{v}_t \hat{\rho}_t) = 0, \qquad \hat{\rho}_{t=0} = \rho_0 \tag{25}$$

*Assume that $\hat{v}_t(x)$ is continuously differentiable in $(t, x)$ and Lipschitz in $x$ uniformily on $(t, x) \in [0, 1] \times \mathbb{R}^d$ with Lipschitz constant $\hat{K}$. Then the square of the $W_2$ distance between $\rho_1$ and $\hat{\rho}_1$ is bounded by*

$$W_2^2(\rho_1, \hat{\rho}_1) \leq e^{1+2\hat{K}} H(\hat{v}) \tag{26}$$

*where $H(\hat{v})$ is the objective function defined in (17).*

The proof of Proposition 3 is given in Appendix E: it leverages the following bound on the square of W-2 distance

$$W_2^2(\rho_1, \hat{\rho}_1) \leq \int_0^1 \int_{\mathbb{R}^d} |X_{t=1}(x) - \hat{X}_{t=1}(x)|^2 \rho_0(x) dx dt \tag{27}$$

where $X_t$ is the flow map solution of (2) with the exact $v_t(x)$ defined in (15) and $\hat{X}_t$ is the flow map obtained by solving (2) with $v_t(x)$ replaced by $\hat{v}_t(x)$.

## 2.2 LINK WITH SCORE-BASED GENERATIVE MODELS

The following result shows that if $\rho_0$ is a Gaussian density, the velocity $v_t(x)$ can be related to the score of the density $\rho_t(x)$:

**Proposition 4.** *Assume that the base density $\rho_0(x)$ is a standard Gaussian density $N(0, \text{Id})$ and suppose that the interpolant $I_t(x_0, x_1)$ is given by (5). Then the score $\nabla \log \rho_t(x)$ is related to velocity $v_t(x)$ as*

$$\nabla \log \rho_t(x) = \begin{cases} -x - \dfrac{2}{\pi} \tan(\tfrac{1}{2}\pi t) v_t(x) & \text{if } t \in [0, 1) \\ -x - \dfrac{4}{\pi^2} \partial_t v_t(x)|_{t=1} & \text{if } t = 1. \end{cases} \tag{28}$$

The proof of this proposition is given in Appendix F. The first formula for $t \in [0, 1)$ is based on a direct calculation using Gaussian integration by parts; the second formula at $t = 1$ is obtained by taking the limit of the first using $v_{t=1}(x) = 0$ from (B.20) and l'Hôpital's rule. It shows that we can in principle resample $\rho_t$ at any $t \in [0, 1]$ using the stochastic differential equation in artificial time $\tau$ whose drift is the score $\hat{s}_t(x)$ obtained by evaluating (28) on the estimated $\hat{v}_t(x)$:

$$dx_\tau = -\hat{s}_t(x_\tau)d\tau + \sqrt{2}dW_\tau. \tag{29}$$

Similarly, the score $\hat{s}_t(x)$ could in principle be used in score-based diffusion models, as explained in Appendix F. We stress however that while our velocity is well-behaved for all times in $[0, 1]$, as shown in (B.20), the drift and diffusion coefficient in the associated SDE are singular at $t = 0, 1$.

## 3 PRACTICAL IMPLEMENTATION AND NUMERICAL EXPERIMENTS

The objective detailed in Section 2 is amenable to efficient empirical estimation, see (I.1), which we utilize to experimentally validate the method. Moreover, it is appealing to consider a neural network parameterization of the velocity field. In this case, the parameters of the model $\hat{v}$ can be optimized through stochastic gradient descent (SGD) or its variants, like Adam (Kingma & Ba, 2015). Following the recent literature regarding density estimation, we benchmark the method on visualizable yet complicated densities that display multimodality, as well as higher dimensional tabular data initially provided in Papamakarios et al. (2017) and tested in other works such as Grathwohl et al. (2019). The 2D test cases demonstrate the ability to flow between empirical densities with *no known analytic* form. In all cases, numerical integration for sampling is done with the Dormand–Prince, explicit Runge-Kutta of order (4)5 (Dormand & Prince, 1980). In Sections 3.1-3.4 the choice of interpolant for experimentation was selected to be that of (5), as it is the one used to draw connections to the technique of score based diffusions in Proposition 4. In Section H we use the interpolant (B.2) and optimize $a_t$ and $b_t$ to investigate the possibility and impact of shortening the path length.

### 3.1 2D DENSITY ESTIMATION

An intuitive first test to benchmark the validity of the method is sampling a target density whose analytic form is known or whose density can be visualized for comparison. To this end, we follow the practice of choosing a few complicated 2-dimensional toy datasets, namely those from (Grathwohl et al., 2019), which were selected to differentiate the flexibility of continuous flows from discrete time flows, which cannot fully separate the modes. We consider anisotropic curved densities, a mixture of 8 separated Gaussians, and a checkerboard density. The velocity field of the interpolant

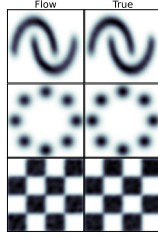 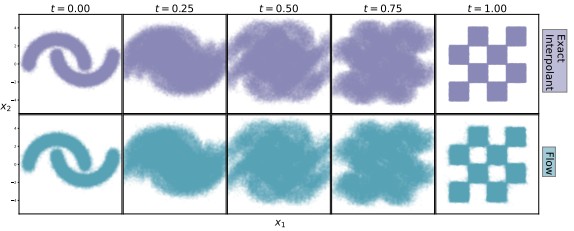

Figure 2: *Left*: 2-D density estimation. *Right*: Learning a flow map between densities when *neither* are analytically known.

| | POWER | GAS | HEPMASS | MINI-BOONE | BSDS300 |
|---|---|---|---|---|---|
| MADE | 3.08 | -3.56 | 20.98 | 15.59 | -148.85 |
| Real NVP | -0.17 | -8.33 | 18.71 | 13.55 | -153.28 |
| Glow | -0.17 | -8.15 | 18.92 | 11.35 | -155.07 |
| CPF | -0.52 | -10.36 | 16.93 | 10.58 | -154.99 |
| NSP | -0.64 | -13.09 | 14.75 | 9.67 | -157.54 |
| FFJORD | -0.46 | -8.59 | 14.92 | 10.43 | -157.40 |
| OT-Flow | -0.30 | -9.20 | 17.32 | 10.55 | -154.20 |
| **Ours** | -0.57 | -12.35 | 14.85 | 10.42 | -156.22 |

| Method | CIFAR-10 | | ImageNet-32x32 | |
|---|---|---|---|---|
| | **NLL** | **FID** | **NLL** | **FID** |
| FFJORD | 3.40 | | | |
| Glow | 3.35 | | 4.09 | |
| DDPM | $\leq 3.75$ | 3.17 | | |
| DDPM++ | $\leq 3.37$ | 2.90 | | |
| ScoreSDE | 2.99 | 2.92 | | |
| VDM | $\leq 2.65$ | 7.41 | $\leq 3.72$ | |
| Soft Truncation | 2.88 | 3.45 | 3.85 | 8.42 |
| ScoreFlow | 2.81 | 5.40 | 3.76 | 10.18 |
| **Ours** | 2.99 | 10.27 | 3.48 | 8.49 |

Table 2: *Left:* Negative log likelihoods (NLL) computed on test data unseen during training (lower is better). Values of MADE, Real NVP, and Glow quoted from the FFJORD paper. Values of OT-Flow, CPF, and NSP quoted from their respective publications. *Right:* NLL and FID scores on unconditional image generation tasks for recent advanced models that emit a likelihood.

flow is parameterized by a simple feed forward neural network with ReLU Nair & Hinton (2010) activation functions. The network for each model has 3 layers, each of width equal to 256 hidden units. Optimizer is performed on $G(\hat{v})$ for 10k epochs. We plot a kernel density estimate over 80k samples from both the flow and true distribution in Figure 2. The interpolant flow captures all the modes of the target density without artificial stretching or smearing, evincing a smooth map.

## 3.2 DATASET INTERPOLATION

As described in Section 2, the velocity field associated to the flow can be inferred from arbitrary densities $\rho_0, \rho_1$ – this deviates from the score-based diffusion perspective, in which one distribution must be taken to be Gaussian for the training paradigm to be tractable.

In Figure 2, we illustrate this capacity by learning the velocity field connecting the anisotropic swirls distribution to that of the checkerboard. The interpolant formulation allows us to draw samples from $\rho_t$ at any time $t \in [0, 1]$, which we exploit to check that the velocity field is empirically correct at all times on the interval, rather than just at the end points. This aspect of interpolants is also noted in (Choi et al., 2022), but for the purpose of density ratio estimation. The above observation highlights an *intrinsic difference* of the proposed method compared to MLE training of flows, where the map that is the minimizer of $G(\hat{v})$ is not empirically known. We stress that query access to $\rho_0$ or $\rho_1$ is not needed to use our interpolation procedure since it only uses samples from these densities.

## 3.3 TABULAR DATA FOR HIGHER DIMENSIONAL TESTING

A set of tabular datasets introduced by (Papamakarios et al., 2017) has served as a consistent test bed for demonstrating flow-based sampling and its associated density estimation capabilities. We continue that practice here to provide a benchmark of the method on models which provide an exact likelihood, separating and comparing to exemplary discrete and continuous flows: MADE (Germain et al., 2015), Real NVP (Dinh et al., 2017), Convex Potential Flows (CPF) (Huang et al., 2021), Neural Spline Flows (NSP) Durkan et al. (2019), Free-form continuous flows (FFJORD) (Grathwohl et al., 2019), and OT-Flow (Finlay et al., 2020). Our primary point of comparison is to other continuous time models, so we sequester them in benchmarking.

We train the interpolant flow model on each target dataset listed in Table 2, choosing the reference distribution of the interpolant $\rho_0$ to be a Gaussian density with mean zero and variance $I_d$, where $d$ is the data dimension. The architectures and hyperparameters are given in Appendix I. We highlight some of the main characteristics of the models here. In each case, sampling of the time $t$ was reweighted according to a beta distribution, with parameters $\alpha, \beta$ provided in the same appendix.

Results from the tabular experiments are displayed in Table 2, in which the negative log-likelihood averaged over a test set of held out data is computed. We note that the interpolant flow achieves better or equivalent held out likelihoods on all ODE based models, except BSDS300, in which the FFJORD outperforms the interpolant by $\sim 0.6\%$. We note upwards of $30\%$ improvements compared to baselines. Note that these likelihoods are achieved *without* direct optimization of it.

### 3.4 Unconditional Image Generation

To compare with recent advances in continuous time generative models such as DDPM (Ho et al., 2020), Score SDE(Song et al., 2021b), and ScoreFlow (Song et al., 2021a), we provide a demonstration of the interpolant flow method on learning to unconditionally generate images trained from the CIFAR-10 (Krizhevsky et al., 2009) and ImageNet 32×32 datasets (Deng et al., 2009; Van Den Oord et al., 2016), which follows suit with ScoreFlow and Variational Diffusion Models (VDM) (Kingma et al., 2021). We train an interpolant flow built from the U-Net architecture from DDPM (Ho et al., 2020) on *a single* NVIDIA A100 GPU, which was previously impossible under maximum likelihood training of continuous time flows. Experimental details can be found in Appendix I. Note that we used a beta distribution reweighting of the time sampling as in the tabular experiments. Table 2 provides a comparison of the negative log likelihoods (NLL), measured in bits per dim (BPD) and Frechet Inception Distance (FID), of our method compared to past flows and state of the art diffusions. We focus our comparison against models which emit a likelihood, as this is necessary to compare NLL.

We compare to other flows FFJORD and Glow (Grathwohl et al., 2019; Kingma & Dhariwal, 2018), as well as to recent advances in score based diffusion in DDPM, DDPM++, VDM, Score SDE, ScoreFlow, and Soft Truncation (Ho et al., 2020; Nichol & Dhariwal, 2021; Kingma et al., 2021; Song et al., 2021b;a; Kim et al., 2022). We present results *without* data augmentation. Our models emit likelihoods, measured in bits per dim, that are competitive with diffusions on both datasets with a NLL of 2.99 and 3.45. Measures of FID are

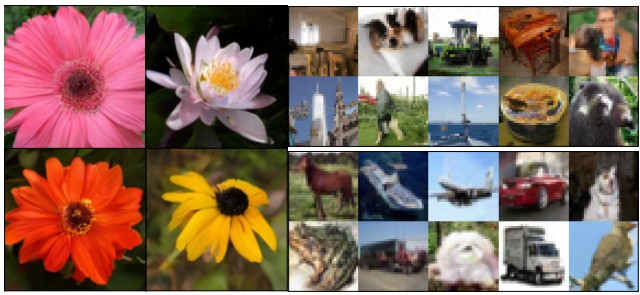

Figure 3: *Left:* InterFlow samples training on 128×128 flowers dataset. *Right:* Samples from flow trained on ImageNet-32×32 (top) and CIFAR-10 (bottom).

proximal to those from diffusions, though slightly behind the best results. We note however that this is a first example of this type of model, and has not been optimized with the training tricks that appear in many of the recent works on diffusions, like exponential moving averages, truncations, learning rate warm-ups, and the like. To demonstrate efficiency on larger domains, we train on the Oxford flowers dataset (Nilsback & Zisserman, 2006), which are images of resolution 128×128. We show example generated images in Figure 3.

## 4 Discussion, Challenges, and Future work

We introduced a continuous time flow method that can be efficiently trained. The approach has a number of intriguing and appealing characteristics. The training circumvents any backpropagation through ODE solves, and emits a stable and interpretable quadratic objective function. This objective has an easily accessible diagnostic which can verify whether a proposed minimizer of the loss is a valid minimizer, and controls the Wasserstein-2 distance between the model and the target.

One salient feature of the proposed method is that choosing an interpolant $I_t(x_0, x_1)$ decouples the optimization problem from that of also choosing a transport path. This separation is also exploited by score-based diffusion models, but our approach offers better explicit control on both. In particular we can interpolate between any two densities in finite time and directly obtain the probability flow needed to calculate the likelihood. Moreover, we showed in Section 2 and Appendices D and G that the interpolant can be optimized to achieve optimal transport, a feature which can reduce the cost of solving the ODE to draw samples. In future work, we will investigate more thoroughly realizations of this procedure by learning the interpolant $I_t$ in a wider class of functions, in addition to minimizing $G(\hat{v})$.

The intrinsic connection to score-based diffusion presented in Proposition 4 may be fruitful ground for understanding the benefits and tradeoffs of SDE vs ODE approaches to generative modeling. Exploring this relation is already underway (Lu et al., 2022; Boffi & Vanden-Eijnden, 2022), and can hopefully provide theoretical insight into designing more effective models.

ACKNOWLEDGMENTS

We thank Gérard Ben Arous, Nick Boffi, Kyle Cranmer, Michael Lindsey, Jonathan Niles-Weed, Esteban Tabak for helpful discussions about transport. MSA is supported by the National Science Foundation under the award PHY-2141336. MSA is grateful for the hospitality of the Center for Computational Quantum Physics at the Flatiron Institute. The Flatiron Institute is a division of the Simons Foundation. EVE is supported by the National Science Foundation under awards DMR-1420073, DMS-2012510, and DMS-2134216, by the Simons Collaboration on Wave Turbulence, Grant No. 617006, and by a Vannevar Bush Faculty Fellowship.

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

## A  BACKGROUND ON TRANSPORT MAPS AND THE CONTINUITY EQUATION

The following result is standard and can be found e.g. in (Villani, 2009; Santambrogio, 2015)

**Proposition A.1.** *Let $\rho_t(x)$ satisfy the continuity equation*

$$\partial_t \rho_t + \nabla \cdot (v_t \rho_t) = 0. \tag{A.1}$$

*Assume that $v_t(x)$ is $C^1$ in both $t$ and $x$ for $t \geq 0$ and globally Lipschitz in $x$. Then, given any $t, t' \geq 0$, the solution of* (A.1) *satisfies*

$$\rho_t(x) = \rho_{t'}(X_{t,t'}(x)) \exp\left(-\int_{t'}^{t} \nabla \cdot v_s(X_{t,s}(x)) ds\right) \tag{A.2}$$

*where $X_{s,t}$ is the probability flow solution to*

$$\frac{d}{dt} X_{s,t}(x) = v_t(X_{s,t}(x)), \qquad X_{s,s}(x) = x. \tag{A.3}$$

*In addition, given any test function $\phi : \Omega \to \mathbb{R}$, we have*

$$\int_{\Omega} \phi(x) \rho_t(x) dx = \int_{\Omega} \phi(X_{t',t}(x)) \rho_{t'}(x) dx. \tag{A.4}$$

In words, Lemma A.1 states that an evaluation of the PDF $\rho_t$ at a given point $x$ may be obtained by evolving the probability flow equation (2) backwards to some earlier time $t'$ to find the point $x'$ that evolves to $x$ at time $t$, assuming that $\rho_{t'}(x')$ is available. In particular, for $t' = 0$, we obtain

$$\rho_t(x) = \rho_0(X_{t,0}(x)) \exp\left(-\int_0^t \nabla \cdot v_s(X_{t,s}(x)) ds\right), \tag{A.5}$$

and

$$\int_{\Omega} \phi(x) \rho_t(x) dx = \int_{\Omega} \phi(X_{0,t}(x)) \rho_0(x) dx. \tag{A.6}$$

*Proof.* The assumed $C^1$ and globally Lipschitz conditions on $v_t$ guarantee global existence (on $t \geq 0$) and uniqueness of the solution to (2). Differentiating $\rho_t(X_{t',t}(x))$ with respect to $t$ and using (2) and (A.1) we deduce

$$\begin{aligned}
\frac{d}{dt} \rho_t(X_{t',t}(x)) &= \partial_t \rho_t(X_{t',t}(x)) + \frac{d}{dt} X_{t',t}(x) \cdot \nabla \rho_t(X_{t',t}(x)) \\
&= \partial_t \rho_t(X_{t',t}(x)) + v_t(X_{t',t}(x)) \cdot \nabla \rho_t(X_{t',t}(x)) \\
&= -\nabla \cdot v_t(X_{t',t}(x)) \, \rho_t(X_{t',t}(x))
\end{aligned} \tag{A.7}$$

Integrating this equation in $t$ from $t = t'$ to $t = t$ gives

$$\rho_t(X_{t',t}(x)) = \rho_{t'}(x) \exp\left(-\int_{t'}^{t} \nabla \cdot v_s(X_{t',s}(x))ds\right) \tag{A.8}$$

Evaluating this expression at $x = X_{t,t'}(x)$ and using the group properties (i) $X_{t',t}(X_{t,t'}(x)) = x$ and (ii) $X_{t',s}(X_{t,t'}(x)) = X_{t,s}(x)$ gives (A.2). Equation (A.4) can be derived by using (A.2) to express $\rho_t(x)$ in the integral at the left hand-side, changing integration variable $x \to X_{t',t}(x)$ and noting that the factor $\exp\left(-\int_{t'}^{t} \nabla \cdot v_s(X_{t,s}(x))ds\right)$ is precisely the Jacobian of this change of variable. The result is the integral at the right hand-side of (A.4). □

## B    PROOF OF PROPOSITION 1

We will work under the following assumption:

**Assumption B.1.** *The densities $\rho_0(x)$ and $\rho_1(x)$ are continuously differentiable in $x$; $I_t(x_0, x_1)$ is continuously differentiable in $(t, x_0, x_1)$ and satisfies (4) and (7); and for all $t \in [0, 1]$ we have*

$$\int_{\mathbb{R}^d} \left| \int_{\mathbb{R}^d \times \mathbb{R}^d} e^{ik \cdot I_t(x_0, x_1)} \rho_0(x_0)\rho_1(x_1)dx_0 dx_1 \right| dk < \infty$$
$$\int_{\mathbb{R}^d} \left| \int_{\mathbb{R}^d \times \mathbb{R}^d} \partial_t I_t(x_0, x_1) e^{ik \cdot I_t(x_0, x_1)} \rho_0(x_0)\rho_1(x_1)dx_0 dx_1 \right| dk < \infty \tag{B.1}$$

This structural assumption guarantees that the stochastic interpolant $x_t$ has a probability density and a probability current. As shown in Appendix C it is satisfied e.g. if $\rho_0$ and $\rho_1$ are Gaussian mixture densities and

$$I_t(x_0, x_1) = a_t x_0 + b_t x_1, \tag{B.2}$$

where $a_t$ and $b_t$ are $C^1$ function of $t \in [0, 1]$ satisfying

$$\dot{a}_t \leq 0, \quad \dot{b}_t \geq 0, \quad a_0 = 1, \quad a_1 = 0, \quad b_0 = 0, \quad b_1 = 1,$$
$$a_t > 0 \text{ on } t \in [0, 1), \quad b_t > 0 \text{ on } t \in (0, 1]. \tag{B.3}$$

The interpolant (4) is in this class for the choice

$$a_t = \cos(\tfrac{1}{2}\pi t), \qquad b_t = \sin(\tfrac{1}{2}\pi t) \tag{B.4}$$

Our proof of Proposition 1 will rely on the following result that quantifies the probability density and the probability current of the stochastic interpolant $x_t$ defined in (6):

**Lemma B.1.** *If Assumption B.1 holds, then the stochastic interpolant $x_t$ defined in (6) has a probability density function $\rho_t(x)$ given by*

$$\rho_t(x) = (2\pi)^{-d} \int_{\mathbb{R}^d \times \mathbb{R}^d \times \mathbb{R}^d} e^{-ik \cdot (x - I_t(x_0, x_1))} \rho_0(x_0)\rho_1(x_1)dx_0 dx_1 dk \tag{B.5}$$

*and it satisfies the continuity equation*

$$\partial_t \rho_t(x) + \nabla \cdot j_t(x) = 0, \qquad \rho_{t=0}(x) = \rho_0(x), \quad \rho_{t=1}(x) = \rho_1(x) \tag{B.6}$$

*withe the probability current $j_t(x)$ given by*

$$j_t(x) = (2\pi)^{-d} \int_{\mathbb{R}^d \times \mathbb{R}^d \times \mathbb{R}^d} \partial_t I_t(x_0, x_1) e^{-ik \cdot (x - I_t(x_0, x_1))} \rho_0(x_0)\rho_1(x_1)dx_0 dx_1 dk \tag{B.7}$$

*In addition the action of $\rho_t(x)$ and $j_t(x)$ against any test function $\phi : \mathbb{R}^d \to \mathbb{R}$ can be expressed as*

$$\int_{\mathbb{R}^d} \phi(x)\rho_t(x)dx = \int_{\mathbb{R}^d \times \mathbb{R}^d} \phi(I_t(x_0, x_1))\rho_0(x_0)\rho_1(x_1)dx_0 dx_1 \tag{B.8}$$

$$\int_{\mathbb{R}^d} \phi(x)j_t(x)dx = \int_{\mathbb{R}^d \times \mathbb{R}^d \times \mathbb{R}^d} \partial_t I_t(x_0, x_1)\phi(I_t(x_0, x_1))\rho_0(x_0)\rho_1(x_1)dx_0 dx_1 \tag{B.9}$$

Note that (B.8) and (B.9) can be formally rewritten as (12) and (14) using the Dirac delta distribution.

*Proof.* By definition of $x_t$ in (6), the characteristic function of this random variable is

$$\mathbb{E}\big[\exp(ik \cdot x_t)\big] = \int_{\mathbb{R}^d \times \mathbb{R}^d} e^{ik \cdot I_t(x_0, x_1)} \rho_0(x_0)\rho_1(x_1)dx_0 dx_1 \tag{B.10}$$

Under Assumption B.1, the Fourier inversion theorem implies that $x_t$ has a density $\rho_t(x)$ given by (B.5). Taking the time derivative of this density gives

$$\begin{aligned}
\partial_t \rho_t(x) &= (2\pi)^{-d} \int_{\mathbb{R}^d \times \mathbb{R}^d \times \mathbb{R}^d} ik \cdot \partial_t I_t(x_0, x_1)e^{-ik \cdot (x - I_t(x_0, x_1))} \rho_0(x_0)\rho_1(x_1)dx_0 dx_1 dk \\
&= -\nabla \cdot j_t(x)
\end{aligned} \tag{B.11}$$

with $j_t(x)$ given by (B.7). $\qquad\square$

Lemma B.1 shows that $\rho_t(x)$ satisfies the continuity equation (3) with the velocity field $v_t(x)$ defined in (15). It also show that the objective function $G(\hat{v})$ in (9) is well-defined, which implies the first part Proposition 1.

For the second part, we will need

**Lemma B.2.** *If Assumption B.1 holds, then*

$$\int_0^1 \int_{\mathbb{R}^d} |v_t(x)|^2 \rho_t(x)dxdt = \mathbb{E}\big[|v_t(I_t(x_0, x_1))|^2\big] \leq \mathbb{E}\big[|\partial_t I_t(x_0, x_1)|^2\big] < \infty \tag{B.12}$$

*Proof.* For $K < \infty$, define

$$\phi_t^K(x) = \begin{cases} 1 & \text{if } |v_t(x)| \leq K \\ 0 & \text{else} \end{cases} \tag{B.13}$$

Then, using the pointwise identity $v_t(x)\rho_t(x) = j_t(x)$ as well as (B.8) and (B.9), we can write

$$\begin{aligned}
0 &= \int_0^1 \int_{\mathbb{R}^d} \phi_t^K(x) \left(2|v_t(x)|^2 \rho_t(x) - 2v_t(x) \cdot j_t(x)\right) dxdt \\
&= 2\mathbb{E}\big[\phi_t^K(I_t)|v_t(I_t)|^2\big] - 2\mathbb{E}\big[\phi_t^K(I_t)\partial_t I_t \cdot v_t(I_t)\big] \\
&= \mathbb{E}\big[\phi_t^K(I_t)|v_t(I_t)|^2\big] + \mathbb{E}\big[\phi_t^K(I_t)|v_t(I_t) - \partial_t I_t|^2\big] - \mathbb{E}\big[\phi_t^K(I_t)|\partial_t I_t|^2\big] \\
&\geq \mathbb{E}\big[\phi_t^K(I_t)|v_t(I_t)|^2\big] - \mathbb{E}\big[\phi_t^K(I_t)|\partial_t I_t|^2\big].
\end{aligned} \tag{B.14}$$

where we use the shorthand $I_t = I_t(x_0, x_1)$ and $\partial_t I_t = \partial_t I_t(x_0, x_1)$. Therefore

$$0 \leq \mathbb{E}\big[\phi_t^K(I_t)|v_t(I_t)|^2\big] \leq \mathbb{E}\big[\phi_t^K(I_t)|\partial_t I_t|^2\big]. \tag{B.15}$$

Since $\lim_{K \to \infty} \mathbb{E}\big[\phi_t^K(I_t)|\partial_t I_t|^2\big] = \mathbb{E}\big[|\partial_t I_t|^2\big]$ and this quantity is finite by assumption we deduce that $\lim_{K \to \infty} \mathbb{E}\big[\phi_t^K(I_t)|v_t(I_t)|^2\big]$ exists and is bounded by $\mathbb{E}\big[|\partial_t I_t|^2\big]$. $\qquad\square$

Lemma B.2 implies that the objective $H(\hat{v})$ in (17) is well-defined, and the second part of statement of Proposition 1 follows from the argument given after (17). For the third part of the proposition we can then proceed as explained in main text, starting from the Poisson equation (B.21).

**Remark B.3.** *Let us show on a simple example that the inequality* $\mathbb{E}\big[|v_t(I_t(x_0, x_1))|^2\big] \leq \mathbb{E}\big[|\partial_t I_t(x_0, x_1)|^2\big]$ *is not saturated in general. Assume that $\rho_0(x)$ is a Gaussian density of mean zero and variance one, and $\rho_1(x)$ a Gaussian density of mean $m \in \mathbb{R}$ and variance one. In this case, if we use the trigonometric interpolant (5), (C.4) indicates that $\rho_t(x)$ is a Gaussian density with mean $\sin(\frac{1}{2}\pi t)m$ and variance one, and (C.6) simplifies to $v_t(x) = \frac{1}{4}\pi^2 m^2 \cos^2(\frac{1}{2}\pi t)$, so that*

$$\int_{\mathbb{R}} |v_t(x)|^2 \rho_t(x)dx = \tfrac{1}{4}\pi^2 m^2 \cos^2(\tfrac{1}{2}\pi t) \tag{B.16}$$

*At the same time*

$$\int_{\mathbb{R}\times\mathbb{R}} |\partial_t I_t(x_0,x_1)|^2 \rho_0(x_0)\rho_1(x_1)dx_0dx_1$$

$$= \tfrac{1}{4}\pi^2 \int_{\mathbb{R}\times\mathbb{R}} \left| -\sin(\tfrac{1}{2}\pi t)x_0 + \cos(\tfrac{1}{2}\pi t)x_1 \right|^2 \rho_0(x_0)\rho_1(x_1)dx_0dx_1$$

$$= \tfrac{1}{4}\pi^2 \left( \sin^2(\tfrac{1}{2}\pi t) + \cos^2(\tfrac{1}{2}\pi t)(1+m^2) \right) \qquad\text{(B.17)}$$

$$= \frac{1}{4}\pi^2 \left( 1 + m^2 \cos^2(\tfrac{1}{2}\pi t) \right)$$

*and so* $\mathbb{E}\big[|v_t(I_t(x_0,x_1))|^2\big] = \mathbb{E}\big[|\partial_t I_t(x_0,x_1))|^2\big] - \tfrac{1}{4}\pi^2 < \mathbb{E}\big[|\partial_t I_t((x_0,x_1))|^2\big].$

The interpolant density $\rho_t(x)$ and the current $j_t(x)$ are given explicitly in Appendix C in the case where $\rho_0$ and $\rho_1$ are both Gaussian mixture densities and we use the linear interpolant (B.2).

Notice that we can evaluate $v_{t=0}(x)$ and $v_{t=1}(x)$ more explicitly. For example, with the linear interpolant (B.2) we have

$$j_{t=0}(x) = \dot{a}_0 x \rho_0(x) + \dot{b}_0 \rho_0(x) \int_{\mathbb{R}^d} x_1 \rho_1(x_1)dx_1,$$

$$j_{t=1}(x) = \dot{b}_1 x \rho_1(x) + \dot{a}_1 \rho_1(x) \int_{\mathbb{R}^d} x_0 \rho_0(x_0)dx_0. \qquad\text{(B.18)}$$

From (12), this implies

$$v_0(x) = \dot{a}_0 x + \dot{b}_0 \int_{\mathbb{R}^d} x_1 \rho_1(x_1)dx_1, \qquad v_1(x) = \dot{b}_1 x + \dot{a}_1 \int_{\mathbb{R}^d} x_0 \rho_0(x_0)dx_0 \qquad\text{(B.19)}$$

For the trigonometric interpolant that uses (B.4) these reduce to

$$v_{t=0}(x) = \tfrac{1}{2}\pi \int_{\mathbb{R}^d} x_1 \rho_1(x_1)dx_1, \qquad v_{t=1}(x) = -\tfrac{1}{2}\pi \int_{\mathbb{R}^d} x_0 \rho_0(x_0)dx_0. \qquad\text{(B.20)}$$

Finally, we note that the result of Proposition 1 remains valid if we work with velocities that are gradient fields. We state this result as:

**Proposition B.4.** *The statements of Proposition 1 hold if $G(\hat{v})$ is minimized over velocities that are gradient fields, in which case the minimizer is of the form $v_t(x) = \nabla\phi_t(x)$ for some potential $\phi_t : \mathbb{R}^d \to \mathbb{R}$ uniquely defined up to a constant.*

Minimizing $G(\hat{v})$ over gradient fields $\hat{v}_t(x) = \nabla\hat{\phi}_t(x)$ guarantees that the minimizer does not contain any component $\tilde{v}_t(x)$ such that $\nabla \cdot (\tilde{v}_t(x)\rho_t(x)) = 0$. Such a component of the velocity has no effect on the evolution of $\rho_t(x)$ but affects the map $X_t$, the solution of (2). We stress however that: *even if we minimize $G(\hat{v})$ over velocities that are not gradient fields, the minimizer $v_t(x)$ produces a map $X_t$ via (2) that satisfies the pushforward condition in (1).*

*Proof.* Define the potential $\phi_t : \mathbb{R}^d \to \mathbb{R}$ as the solution to

$$\nabla \cdot (\rho_t \nabla\phi_t) = \nabla \cdot j_t \qquad\text{(B.21)}$$

with $\rho_t(x)$ and $j_t(x)$ given by (12) and (14), respectively. This is a Poisson equation for $\phi_t(x)$ which has a unique (up to a constant) solution by the Fredholm alternative since $\rho_t(x)$ and $j_t(x)$ have the same support and $\int_{\mathbb{R}^d} \nabla \cdot j_t(x)dx = 0$. In terms of $\phi_t(x)$, (13) can therefore be written as

$$\partial_t \rho_t + \nabla \cdot (\rho_t \nabla\phi_t) = 0 \qquad\text{(B.22)}$$

The velocity $\nabla\phi_t(x)$ is also the unique minimizer of the objective (9) over gradient fields since if we set $\hat{v}_t(x) = \nabla\hat{\phi}_t(x)$ and optimize $G(\nabla\hat{\phi})$ over $\hat{\phi}$, the Euler-Lagrange equation for the minimizer is precisely the Poisson equation (B.21). The lower bound on the objective evaluated at $\nabla\phi_t(x)$ still holds since in the argument above involving (17) and (18) can be made by replacing the identity $v_t(x)\rho_t(x) = j_t(x)$ with $\int_{\mathbb{R}^d} \nabla\hat{\phi}_t(x) \cdot \nabla\phi_t(x)\rho_t(x)dx = \int_{\mathbb{R}^d} \nabla\hat{\phi}_t(x) \cdot j_t(x)dx$, which is (B.21) written in weak form. $\qquad\square$

Interestingly, with $v_t(x)$ defined in (15) and $\phi_t(x)$ defined as the solution to (B.21), we have

$$\int_{\mathbb{R}^d} |v_t(x)|^2 \rho_t(x) dx \geq \int_{\mathbb{R}^d} |\nabla \phi_t(x)|^2 \rho_t(x) dx \tag{B.23}$$

consistent with the fact that the gradient field $\nabla \phi_t(x)$ is the velocity that minimizes $\int_{\mathbb{R}^d} |\hat{v}_t(x)|^2 \rho_t(x) dx$ over all $\hat{v}_t(x)$ such that $\partial_t \rho_t + \nabla \cdot (\hat{v}_t \rho_t) = 0$ with $\rho_t(x)$ given by (12). We stress however that, even if we work we gradient fields, the inequality (21) is not saturated in general.

**Remark B.5.** *If we assume that $\hat{v}_t(x) = \nabla \hat{\phi}_t(x)$, we can write the objective in (16) as*

$$\begin{aligned}
G(\nabla \hat{\phi}) &= \int_0^1 \int_{\mathbb{R}^d} \left( |\nabla \hat{\phi}_t(x)|^2 \rho_t(x) - 2\nabla \hat{\phi}_t(x) \cdot j_t(x) \right) dx dt \\
&= \int_0^1 \int_{\mathbb{R}^d} \left( |\nabla \hat{\phi}_t(x)|^2 \rho_t(x) + 2\hat{\phi}_t(x) \nabla \cdot j_t(x) \right) dx dt \\
&= \int_0^1 \int_{\mathbb{R}^d} \left( |\nabla \hat{\phi}_t(x)|^2 \rho_t(x) - 2\hat{\phi}_t(x) \partial_t \rho_t(x) \right) dx dt \\
&= \int_0^1 \int_{\mathbb{R}^d} \left( |\nabla \hat{\phi}_t(x)|^2 \rho_t(x) + 2\partial_t \hat{\phi}_t(x) \rho_t(x) \right) dx dt \\
&\quad + 2 \int_{\mathbb{R}^d} \left( \hat{\phi}_{t=0}(x) \rho_0(x) - \hat{\phi}_{t=1}(x) \rho_1(x) \right) dx,
\end{aligned} \tag{B.24}$$

*where we integrated by parts in $x$ to get the second equality, we used the continuity equation (13) to get the second, and integrated by parts in $t$ to get the third using $\rho_{t=0} = \rho_0$ and $\rho_{t=1} = \rho_1$. Since the objective in the last expression is an expectation with respect to $\rho_t$, we can evaluate it as*

$$G(\nabla \hat{\phi}) = \mathbb{E}\left( |\nabla \hat{\phi}_t(I_t)|^2 + 2\partial_t \hat{\phi}_t(I_t) \right) + 2\mathbb{E}_0 \hat{\phi}_{t=0} - 2\mathbb{E}_1 \hat{\phi}_{t=1}, \tag{B.25}$$

*where $\mathbb{E}_1$ and $\mathbb{E}_0$ denote expectations with respect to $\rho_1$ and $\rho_0$, respectively. Therefore, we could use (B.25) as alternative objective to obtain $v_t(x) = \nabla \phi_t(x)$ by minimization. This objective is, up to a sign and a factor 2, the KILBO objective introduced in Neklyudov et al. (2022). Notice that using the original objective in (9) avoids the computation of derivatives in $x$ and $t$, and allows one to work with $\hat{v}_t(x)$ directly.*

## C  THE CASE OF GAUSSIAN MIXTURE DENSITIES

Here we consider the case where $\rho_0$ and $\rho_1$ are both Gaussian mixture densities. We denote

$$\begin{aligned}
N(x|m, C) &= (2\pi)^{-d/2} [\det C]^{-1/2} \exp\left( -\tfrac{1}{2}(x-m)^T C^{-1}(x-m) \right) \\
&= (2\pi)^{-d} \int_{\mathbb{R}^d} e^{ik \cdot (x-m) - \frac{1}{2} k^T C k} dk
\end{aligned} \tag{C.1}$$

the Gaussian probability density with mean vector $m \in \mathbb{R}^d$ and positive-definite symmetric covariance matrix $C = C^T \in \mathbb{R}^d \times \mathbb{R}^d$. We assume that

$$\rho_0(x) = \sum_{i=1}^{N_0} p_i^0 N(x|m_i^0, C_i^0), \qquad \rho_1(x) = \sum_{i=1}^{N_1} p_i^1 N(x|m_i^1, C_i^1) \tag{C.2}$$

where $N_0, N_1 \in \mathbb{N}$, $p_i^0 > 0$ with $\sum_{i=1}^{N_0} p_i^0 = 1$, $m_i^0 \in \mathbb{R}^d$, $C_i^0 = (C_i^0)^T \in \mathbb{R}^d \times \mathbb{R}^d$, positive-definite, and similarly for $p_i^1$, $m_i^1$, and $C_i^1$. We assume that the interpolant is of the form (B.2) and we denote

$$m_t^{ij} = a_t m_i^0 + b_t m_j^1, \qquad C_t^{ij} = a_t^2 C_i^0 + b_t^2 C_j^1, \qquad i = 1, \ldots, N_0, \quad j = 1, \ldots, N_1 \tag{C.3}$$

Note that if all the covariance matrices are the same, $C_i^0 = C_j^1 = C$, with the trigonometric interpolant in (5) we have $C_t^{ij} = C$, which justifies this choice of interpolant.

We have:

**Proposition C.1.** *The interpolant density $\rho_t(x)$ obtained by connecting the probability densities in* (C.2) *using the linear interpolant* (B.2) *is given by*

$$\rho_t(x) = \sum_{i=1}^{N_0} \sum_{j=1}^{N_1} p_i^0 p_j^1 N(x|m_t^{ij}, C_t^{ij}) \tag{C.4}$$

*and it satisfies the continuity equation $\partial_t \rho_t(x) + \nabla \cdot j_t(x) = 0$ with the current*

$$j_t(x) = \sum_{i=1}^{N_0} \sum_{j=1}^{N_1} p_i^0 p_j^1 \left( \dot{m}_t^{ij} + \tfrac{1}{2} \dot{C}_t^{ij} (C_t^{ij})^{-1} (x - m_t^{ij}) \right) N(x|m_t^{ij}, C_t^{ij}) \tag{C.5}$$

This proposition implies that

$$v_t(x) = \frac{\sum_{i=1}^{N_0} \sum_{j=1}^{N_1} p_i^0 p_j^1 \left( \dot{m}_t^{ij} + \tfrac{1}{2} \dot{C}_t^{ij} (C_t^{ij})^{-1} (x - m_t^{ij}) \right) N(x|m_t^{ij}, C_t^{ij})}{\sum_{i=1}^{N_0} \sum_{j=1}^{N_1} p_i^0 p_j^1 N(x|m_t^{ij}, C_t^{ij})} \tag{C.6}$$

This velocity field is growing at most linearly in $x$, and when the mode of the Gaussian are well separated, in each mode it approximately reduces to

$$\dot{m}_t^{ij} + \tfrac{1}{2} \dot{C}_t^{ij} (C_t^{ij})^{-1} (x - m_t^{ij}) \tag{C.7}$$

*Proof.* Using the Fourier representation in (C.1) and proceeding as in the proof of Lemma B.1, we deduce that $\rho_t(x)$ is given by

$$\rho_t(x) = (2\pi)^{-d} \sum_{i=1}^{N_0} \sum_{j=1}^{N_1} p_i^0 p_j^1 \int_{\mathbb{R}^d} e^{ik \cdot (x - m_t^{ij}) - \frac{1}{2} k^T C_t^{ij} k} dk. \tag{C.8}$$

Performing the integral over $k$ gives (C.4). Taking the time derivative of this density gives

$$
\begin{aligned}
\partial_t \rho_t(x) &= -(2\pi)^{-d} \sum_{i=1}^{N_0} \sum_{j=1}^{N_1} p_i^0 p_j^1 \int_{\mathbb{R}^d} \left( ik \cdot \dot{m}_t^{ij} + \tfrac{1}{2} k^T \dot{C}_t^{ij} k \right) e^{ik \cdot (x - m_t^{ij}) - \frac{1}{2} k^T C_t^{ij} k} dk \\
&= -(2\pi)^{-d} \sum_{i=1}^{N_0} \sum_{j=1}^{N_1} p_i^0 p_j^1 \int_{\mathbb{R}^d} ik \cdot \left( \dot{m}_t^{ij} - \tfrac{1}{2} i \dot{C}_t^{ij} k \right) e^{ik \cdot (x - m_t^{ij}) - \frac{1}{2} k^T C_t^{ij} k} dk \\
&= -\nabla \cdot j_t(x)
\end{aligned} \tag{C.9}
$$

with

$$
\begin{aligned}
j_t(x) &= (2\pi)^{-d} \sum_{i=1}^{N_0} \sum_{j=1}^{N_1} p_i^0 p_j^1 \int_{\mathbb{R}^d} (\dot{m}_t^{ij} - \tfrac{1}{2} i \dot{C}_t^{ij} k) e^{ik \cdot (x - m_t^{ij}) - \frac{1}{2} k^T C_t^{ij} k} dk \\
&= (2\pi)^{-d} \sum_{i=1}^{N_0} \sum_{j=1}^{N_1} p_i^0 p_j^1 \dot{m}_t^{ij} \int_{\mathbb{R}^d} e^{ik \cdot (x - m_t^{ij}) - \frac{1}{2} k^T C_t^{ij} k} dk \\
&\quad - \tfrac{1}{2} (2\pi)^{-d} \sum_{i=1}^{N_0} \sum_{j=1}^{N_1} p_i^0 p_j^1 \dot{C}_t^{ij} \nabla \int_{\mathbb{R}^d} e^{ik \cdot (x - m_t^{ij}) - \frac{1}{2} k^T C_t^{ij} k} dk \\
&= \sum_{i=1}^{N_0} \sum_{j=1}^{N_1} p_i^0 p_j^1 \left( \dot{m}_t^{ij} - \tfrac{1}{2} \dot{C}_t^{ij} \nabla \right) N(x|m_t^{ij}, C_t^{ij}) \\
&= \sum_{i=1}^{N_0} \sum_{j=1}^{N_1} p_i^0 p_j^1 \left( \dot{m}_t^{ij} + \tfrac{1}{2} \dot{C}_t^{ij} (C_t^{ij})^{-1} (x - m_t^{ij}) \right) N(x|m_t^{ij}, C_t^{ij})
\end{aligned} \tag{C.10}
$$

$\square$

## D  OPTIMIZING TRANSPORT THROUGH STOCHASTIC INTERPOLANTS

Using the velocity $v_t(x)$ in (15) that minimizes the objective (9) in the ODE (2) gives an exact transport map $T = X_{t=1}$ from $\rho_0$ to $\rho_1$. However, this map is not optimal in general, in the sense that it does not minimize

$$\int_{\mathbb{R}^d} |\hat{T}(x) - x|^2 \rho_0(x) dx \tag{D.1}$$

over all $\hat{T}$ such that $\hat{T} \sharp \rho_0 = \rho_1$. It is easy to understand why: In their seminal paper Benamou & Brenier (2000) showed that finding the optimal map requires solving the minimization problem

$$\min_{(\hat{v}, \hat{\rho})} \int_0^1 \int_{\mathbb{R}^d} |\hat{v}_t(x)|^2 \hat{\rho}_t(x) dx dt \tag{D.2}$$
$$\text{subject to:} \quad \partial_t \hat{\rho}_t + \nabla \cdot (\hat{v}_t \hat{\rho}_t) = 0, \quad \hat{\rho}_{t=0} = \rho_0, \quad \hat{\rho}_{t=1} = \rho_1.$$

As also shown in (Benamou & Brenier, 2000), the velocity minimizing (D.2) is a gradient field, $v_t^*(x) = \nabla \phi_t^*(x)$, and the minimizing couple $(\rho_t^*, \phi_t^*)$ is unique and satisfies

$$\partial_t \rho_t^* + \nabla \cdot (\nabla \phi_t^* \rho_t^*) = 0, \quad \rho_{t=0}^* = \rho_0, \quad \rho_{t=1}^* = \rho_1 \tag{D.3}$$
$$\partial_t \phi_t^* + \tfrac{1}{2} |\nabla \phi_t^*|^2 = 0.$$

In contrast, in our construction the interpolant density $\rho_t(x)$ is fixed by the choice of interpolant $I_t(x_0, x_1)$, and $\rho_t(x) \neq \rho_t^*(x)$ in general. As a result, the value of $\int_0^1 \int_{\mathbb{R}^d} |v_t(x)|^2 \rho_t(x) dx dt = \mathbb{E}[|v_t(I_t)|^2]$ for the velocity $v_t(x)$ minimizing (9) is not the minimum in (D.2).

Minimizing (9) over gradient fields reduces the value of the objective in (D.2), but it does not yield an optimal map either—indeed the gradient velocity $\nabla \phi_t(x)$ with the potential $\phi_t(x)$ solution to (B.21) only minimizes the objective in (D.2) over all $\hat{v}_t(x)$ with $\hat{\rho}_t(x) = \rho_t(x)$ fixed, as explained after (B.23).

It is natural to ask whether our stochastic interpolant construction can be amended or generalized to derive optimal maps. This question is discussed next, from two complementary perspectives: via optimization of the interpolant $I_t(x_0, x_1)$, and/or via optimization of the base density $\rho_0$, assuming that we have some leeway to choose this density.

### D.1  OPTIMAL TRANSPORT WITH OPTIMAL INTERPOLANTS

Since (10) indicates that the minimum of $G(\hat{v})$ is lower bounded by minus the value of the objective in (D.2), one way to optimize the transport is to maximize this minimum over the interpolant. Under some assumption on the Benamou-Brenier density $\rho_t^*(x)$ solution of (D.3), this procedure works, as we show now. Let us begin with a definition:

**Definition D.1** (Interpolable density). *We say that one-parameter family of probability densities $\rho_t(x)$ with $t \in [0, 1]$ is interpolable (in short: $\rho_t(x)$ is interpolable) if there exists a one-parameter family of invertible maps $T_t : \mathbb{R}^d \to \mathbb{R}^d$ with $t \in [0, 1]$, continuously-differentiable in time and space, such that $\rho_t$ is the pushforward by $T_t$ of the Gaussian density with mean zero and covariance identity, i.e. $T_t \sharp N(0, Id) = \rho_t$ for all $t \in [0, 1]$.*

Interpolable densities form a wide class, as discussed e.g. in Mikulincer & Shenfeld (2022). These densities also are the ones that can be learned by score-based diffusion modeling discussed in Section 2.2. They are useful for our purpose because of the following result showing that any interpolable density can be represented as an interpolant density:

**Proposition D.1.** *Let $\rho_t(x)$ be an interpolable density in the sense of Definition D.1. Then*

$$I_t(x_0, x_1) = T_t \big( T_0^{-1}(x_0) \cos(\tfrac{1}{2}\pi t) + T_1^{-1}(x_1) \sin(\tfrac{1}{2}\pi t) \big) \tag{D.4}$$

*satisfies* (4) *and is such that the stochastic interpolant defined in* (6) *satisfies* $x_t \sim \rho_t$.

We stress that the interpolant in (D.4) is in general not the only one giving the interpolable density $\rho_t(x)$, and the actual value of the map $T_t$ plays no role in the results below.

*Proof.* First notice that

$$I_{t=0}(x_0, x_1) = T_0(T_0^{-1}(x_0)) = x_0, \qquad I_{t=1}(x_0, x_1) = T_1(T_1^{-1}(x_0)) = x_1 \tag{D.5}$$

so that the boundary condition in (4) are satisfied. Then observe that, by definition of $T_t$,

$$\begin{aligned} x_0 \sim \rho_0 &\quad \Rightarrow \quad T_0^{-1}(x_0) \sim N(0, \text{Id}), \\ x_1 \sim \rho_1 &\quad \Rightarrow \quad T_1^{-1}(x_1) \sim N(0, \text{Id}), \end{aligned} \tag{D.6}$$

This implies that, if $x_0 \sim \rho_0$, $x_1 \sim \rho_1$, and they are independent,

$$T_0^{-1}(x_0) \cos(\tfrac{1}{2}\pi t) + T_1^{-1}(x_1) \sin(\tfrac{1}{2}\pi t) \tag{D.7}$$

is a Gaussian random variable with mean zero and covariance

$$\text{Id} \cos^2(\tfrac{1}{2}\pi t) + \text{Id} \sin^2(\tfrac{1}{2}\pi t) = \text{Id}, \tag{D.8}$$

i.e. it is a sample from $N(0, \text{Id})$. By definition of $T_t$, this implies that

$$x_t = I_t(x_0, x_1) = T_t\big(T_0^{-1}(x_0) \cos(\tfrac{1}{2}\pi t) + T_1^{-1}(x_1) \sin(\tfrac{1}{2}\pi t)\big) \tag{D.9}$$

is a sample from $\rho_t$ if $x_0 \sim \rho_0$, $x_1 \sim \rho_1$, and they are independent. $\qquad\square$

Proposition D.1 implies that:

**Proposition D.2.** *Assume that (i) the optimal density function $\rho_t^*(x)$ minimizing* (D.2) *is interpolable and (ii)* (D.3) *has classical solution. Consider the max-min problem*

$$\max_{\hat{I}} \min_{\hat{v}} G(\hat{v}) \tag{D.10}$$

*where $G(\hat{v})$ is the objective in* (9) *and the maximum is taken over interpolants satisfying* (4)*. Then a maximizer of* (D.10) *exists, and any maximizer $I_t^*(x_0, x_1)$ is such that the probability density function of $x_t^* = I_t^*(x_0, x_1)$, with $x_0 \sim \rho_0$ and $x_1 \sim \rho_1$ independent, is the optimal $\rho_t^*(x)$, the mimimizing velocity is $v_t^*(x) = \nabla\phi_t^*(x)$, and the pair $(\rho_t^*(x), \phi_t^*(x))$ satisfies* (D.3)*.*

The proof of Proposition D.2 relies on the following simple reformulation of (D.2):

**Lemma D.3.** *The Benamou-Brenier minimization problem in* (D.2) *is equivalent to the min-max problem*

$$\max_{\hat{\rho}, \hat{j}} \min_{\hat{v}} \int_0^1 \int_{\mathbb{R}^d} \big(\tfrac{1}{2}|\hat{v}_t(x)|^2 \hat{\rho}_t(x) - \hat{v}_t(x) \cdot \hat{j}_t(x)\big) \, dx dt$$
$$= \min_{\hat{v}} \max_{\hat{\rho}, \hat{j}} \int_0^1 \int_{\mathbb{R}^d} \big(\tfrac{1}{2}|\hat{v}_t(x)|^2 \hat{\rho}_t(x) - \hat{v}_t(x) \cdot \hat{j}_t(x)\big) \, dx dt \tag{D.11}$$
$$\textit{subject to:} \quad \partial_t \hat{\rho}_t + \nabla \cdot \hat{j}_t = 0, \quad \hat{\rho}_{t=0} = \rho_0, \quad \hat{\rho}_{t=1} = \rho_1$$

*In particular, under the conditions on $\rho_0$ and $\rho_1$ such that* (D.2) *has a minimizer, the optimizer $(\rho_t^*, v_t^*, j_t^*)$ is unique and satisfies $v_t^*(x) = \nabla\phi_t^*(x)$, $j_t^*(x) = \nabla\phi_t^*(x)\rho_t^*(x)$ with $(\rho_t^*, \phi_t^*)$ solution to* (D.3)*.*

*Proof.* Since (D.11) is convex in $\hat{v}$ and concave in $(\hat{\rho}, \hat{j})$, the min-max and the max-min are equivalent by von Neumann's minimax theorem. Considering the problem where we minimize over $\hat{v}_t(x)$ first, the minimizer must satisfy:

$$\hat{v}_t(x)\hat{\rho}_t(x) = \hat{j}_t(x) \tag{D.12}$$

Since $\hat{\rho}_t(x) \geq 0$ and $\hat{j}_t(x)$ have the same support by the constraint in (D.11), the solution to this equation is unique on this support. Using (D.12) in (D.11), we can therefore rewrite the max-min problem as

$$\max_{\hat{\rho}} -\tfrac{1}{2} \int_0^1 \int_{\mathbb{R}^d} |\hat{v}_t(x)|^2 \hat{\rho}_t(x) dx dt$$
$$\textit{subject to:} \quad \partial_t \hat{\rho}_t + \nabla \cdot \big(\hat{v}_t \rho_t\big) = 0, \quad \hat{\rho}_{t=0} = \rho_0, \quad \hat{\rho}_{t=1} = \rho_1 \tag{D.13}$$

This problem is equivalent to the Benamou-Brenier minimization problem in (D.2).

To write the Euler-Lagrange equations for the optimizers of the min-max problem (D.11), let us introduce the extended objective

$$\int_0^1 \int_{\mathbb{R}^d} \left(\tfrac{1}{2}|\hat{v}_t(x)|^2 \hat{\rho}_t(x) - \hat{v}_t(x) \cdot \hat{\jmath}_t(x)\right) dxdt - \int_0^1 \int_{\mathbb{R}^d} \hat{\phi}_t(x)\left(\partial_t \hat{\rho}_t + \nabla \cdot \hat{\jmath}_t(x)\right) dxdt$$
$$+ \int_{\mathbb{R}^d} (\hat{\phi}_1(x)\rho_1 - \hat{\phi}_0(x)\rho_0)dx \tag{D.14}$$

where $\hat{\phi}_t : \mathbb{R}^d \to \mathbb{R}$ is a Lagrangian multiplier to be determined. The Euler-Lagrange equations can be obtained by taking the first variation of the objective (D.14) over $\hat{\phi}$, $\hat{\rho}$, $\hat{\jmath}$, and $\hat{v}$, respectively. They read

$$\begin{aligned} 0 &= \partial_t \rho_t^* + \nabla \cdot j_t^*, \quad \rho_{t=0}^* = \rho_0, \quad \rho_{t=1}^* = \rho_1 \\ 0 &= \partial_t \phi_t^* + \tfrac{1}{2}|v_t^*|^2, \\ 0 &= -v_t^* + \nabla \phi_t^*, \\ 0 &= v_t^* \rho_t^* - j_t^*. \end{aligned} \tag{D.15}$$

These equations imply that $v_t^*(x) = \nabla \phi_t^*(x)$, $j_t^*(x) = v_t^*(x)\rho_t^*(x) = \nabla \phi_t^*(x)\rho_t^*(x)$, with $(\rho_t^*, \phi_t^*)$ solution to (D.3), as stated in the lemma. Since the optimization problem in (D.11) is convex in $\hat{v}$ and concave in $(\hat{\rho}, \hat{\jmath})$, its optimizer is unique and solves these equations. $\qquad\square$

*Proof of Proposition D.2.* We can reformulate the max-min problem (D.10) as

$$\max_{\hat{\rho},\hat{\jmath}} \min_{\hat{v}} \int_0^1 \int_{\mathbb{R}^d} \left(\tfrac{1}{2}|\hat{v}_t(x)|^2 \hat{\rho}_t(x) - \hat{v}_t(x) \cdot \hat{\jmath}_t(x)\right) dxdt \tag{D.16}$$

where the maximization is taken over probability density functions $\hat{\rho}_t(x)$ and probability currents $\hat{\jmath}_t(x)$ as in (B.5) and (B.7) with $I_t$ replaced by $\hat{I}_t$, i.e. formally given in terms of the Dirac delta distribution as

$$\hat{\rho}_t(x) = \int_{\mathbb{R}^d \times \mathbb{R}^d} \delta\left(x - \hat{I}_t(x_0, x_1)\right)\rho_0(x_0)\rho_1(x_1)dx_0 dx_1$$
$$\hat{\jmath}_t(x) = \int_{\mathbb{R}^d \times \mathbb{R}^d} \partial_t \hat{I}_t(x_0, x_1)\delta\left(x - \hat{I}_t(x_0, x_1)\right)\rho_0(x_0)\rho_1(x_1)dx_0 dx_1 \tag{D.17}$$

Since this pair automatically satisfies

$$\partial_t \hat{\rho}_t + \nabla \cdot \hat{\jmath}_t = 0, \quad \hat{\rho}_{t=0} = \rho_0, \quad \hat{\rho}_{t=1} = \rho_1, \tag{D.18}$$

the max-min problem (D.16) is similar to the one considered in Lemma (D.3), except that the maximization is taken over probability density functions and associated currents that can be written as in (D.17). By Proposition D.1, this class is large enough to represent $\rho_t^*(x)$ since we have assumed that $\rho_t^*(x)$ is an interpolable density, and the statement of the proposition follows. $\qquad\square$

Since our primary aim here is to construct a map $T = X_{t=1}$ that pushes forward $\rho_0$ onto $\rho_1$, but not necessarily to identify the optimal one, we can perform the maximization over $\hat{I}_t(x_0, x_1)$ in a restricted class (though of course the corresponding map is no longer optimal in that case). We investigate this option in numerical examples in Appendix H, using interpolants of the type (B.2) and maximizing over the functions $a_t$ and $b_t$, subject to $a_0 = b_1 = 1$, $a_1 = b_0 = 0$. In Section G we also discussion generalizations of the interpolant that can render the optimization of the transport easier to perform. We leave the full investigation of the consequences of Proposition D.2 for future work.

**Remark D.4** (Optimal interpolant for Gaussian densities). *Note that if $\rho_0$ and $\rho_1$ are both Gaussian densities with respective mean $m_0 \in \mathbb{R}^d$ and $m_1 \in \mathbb{R}^d$ and the same covariance $C \in \mathbb{R}^d \times \mathbb{R}^d$, an interpolant of the type D.4 is*

$$I_t(x_0, x_1) = \cos(\tfrac{1}{2}\pi t)(x_0 - m_0) + \sin(\tfrac{1}{2}\pi t)(x_1 - m_1 1) + (1 - t)m_0 + tm_1, \tag{D.19}$$

*and a calculation similar to the one presented in Appendix C shows that the associated velocity field $v_t(x)$ is*

$$v_t(x) = (m_1 - m_0) \tag{D.20}$$

*This is the velocity giving the optimal transport map $X_t(x) = x + (m_1 - m_0)t$.*

**Remark D.5** (Rectifying the map). *In (Liu et al., 2022; Liu, 2022), where an approach similar to ours using the linear interpolant $x_t = x_0(1 - t) + x_1 t$ is introduced, an alternative procedure is proposed to optimize the transport. Specifically, it is suggested to rectify the map $T = X_{t=1}$ learned by repeating the procedure using the new interpolant $x'_t = x_0(1 - t) + T(x_0)t$ with $x_0 \sim \rho_0$. As shown in (Liu, 2022) iterating on this rectification step yields successive maps that are getting closer to optimality. The main drawback of this approach is that it requires each of these maps (including the first one) to be learned exactly, i.e. we must have $T\sharp\rho_0 = \rho_1$ to use the interpolant $x'_t$ above. If the maps are not exact, which is unavoidable in practice, the procedure introduces a bias whose amplitude will grow with the iterations, leading to instability (as noted in Liu et al. (2022); Liu (2022)).*

## D.2 OPTIMIZING THE BASE DENSITY

While our construction allows one to connect any pair of densities $\rho_0$ and $\rho_1$, the typical situation of interest is when $\rho_1$ is a complex target density and we wish to construct a generative model for this density by transporting samples from a simple base density $\rho_0$. In this case it is natural to adjust parameters in $\rho_0$ to optimize the transport via maximization of $G(v) = \min_{\hat{v}} G(\hat{v})$ over these parameters—indeed changing $\rho_0$ also affects the stochastic interpolant $x_t$ defined in (6), and hence both the interpolant density $\rho_t(x)$ and the velocity $v_t(x)$. For example we can take $\rho_0$ to be a Gaussian with mean $m \in \mathbb{R}^d$ and covariance $C \in \mathbb{R}^d \times \mathbb{R}^d$, and maximize $G(v) = \min_{\hat{v}} G(\hat{v})$ over $m$ and $C$. This construction is tested in the numerical examples treated in Appendix H. We also discuss how to generalize it in Section G.

Optimizing $\rho_0$ only makes practical sense if we do so in a restricted class, like that of Gaussian densities that we just discussed. Still, we may wonder whether optimizing $\rho_0$ over all densities would automatically give $\rho_0 = \rho_1$ and $v_t = 0$. If the interpolant is fixed, the answer is no, in general. Indeed, even if we set $\rho_0 = \rho_1$, the interpolant density will still evolve, i.e. $\rho_t \neq \rho_0$ except at $t = 0, 1$, in general. This indicates that optimizing the interpolant in concert with $\rho_0$ is necessary if we want to optimize the transport.

## E PROOF OF PROPOSITION 3

To use the bound in (27), let us consider the evolution of

$$Q_t = \int_{\mathbb{R}^d} |X_t(x) - \hat{X}_t(x)|^2 \rho_0(x) dx \tag{E.1}$$

Using $\dot{X}_t(x) = v_t(X_t(x))$ and $\dot{\hat{X}}_t(x) = \hat{v}_t(\hat{X}_t(x))$, we deduce

$$\dot{Q}_t = 2 \int_{\mathbb{R}^d} (X_t(x) - \hat{X}_t(x)) \cdot (v_t(X_t(x)) - \hat{v}_t(\hat{X}_t(x)))\rho_0(x)dx$$

$$= 2 \int_{\mathbb{R}^d} (X_t(x) - \hat{X}_t(x)) \cdot (v_t(X_t(x)) - \hat{v}_t(X_t(x)))\rho_0(x)dx \tag{E.2}$$

$$+ 2 \int_{\mathbb{R}^d} (X_t(x) - \hat{X}_t(x)) \cdot (\hat{v}_t(X_t(x)) - \hat{v}_t(\hat{X}_t(x)))\rho_0(x)dx$$

Now use

$$2(X_t - \hat{X}_t) \cdot (v_t(X_t) - \hat{v}_t(X_t)) \leq |X_t - \hat{X}_t|^2 + |v_t(X_t) - \hat{v}_t(X_t)|^2 \tag{E.3}$$

and

$$2(X_t - \hat{X}_t) \cdot (\hat{v}_t(X_t) - \hat{v}_t(\hat{X}_t)) \leq 2\hat{K}|X_t - \hat{X}_t|^2 \tag{E.4}$$

to obtain

$$\dot{Q}_t \leq (1 + 2\hat{K})Q_t + \int_{\mathbb{R}^d} |v_t(X_t(x)) - \hat{v}_t(X_t(x))|^2 \rho_0(x)dx \tag{E.5}$$

Therefore, by Gronwall's inequality and since $Q_0 = 0$ we deduce

$$Q_1 \leq e^{1+2\hat{K}} \int_0^1 \int_{\mathbb{R}^d} |v_t(X_t(x)) - \hat{v}_t(X_t(x))|^2 \rho_0(x)dxdt = e^{1+2\hat{K}} H(\hat{v}). \tag{E.6}$$

Since $W_2^2(\rho_1, \hat{\rho}_1) \leq Q_1$ by (27), we are done. $\qquad \square$

Note that the proposition suggests to regularize $G(\hat{v})$ using e.g.

$$G_\lambda(\hat{v}) = G(\hat{v}) + \lambda \int_0^1 \int_{\mathbb{R}^d} \|\nabla \hat{v}_t(x)\|^2 \rho_t(x) dx dt = G(\hat{v}) + \lambda \mathbb{E}\left[\|\nabla \hat{v}_t(I_t(x_0, x_1)\|^2\right] \quad \text{(E.7)}$$

with some small $\lambda > 0$. In the numerical results presented in the paper no such regularization was included.

## F    PROOF OF PROPOSITION 4 AND LINK WITH SCORE-BASED DIFFUSION MODELS

Assume that the interpolant is of the type (B.2) so that $\partial_t I_t(x_0, x_1) = \dot{a}_t x_0 + \dot{b}_t x_1$. For $t \in (0, 1)$ let us write expression (14) for the probability current as

$$
\begin{aligned}
j_t(x) &= \int_{\mathbb{R}^d \times \mathbb{R}^d} (\dot{a}_t x_0 + \dot{b}_t x_1) \delta(x - a_t x_0 + b_t x_1) \rho_0(x_0) \rho_1(x_1) dx_0 dx_1 \\
&= \int_{\mathbb{R}^d \times \mathbb{R}^d} \left( \frac{\dot{b}_t}{b_t}(a_t x_0 + b_t x_1) + \left(\dot{a}_t - \frac{\dot{b}_t}{b_t} a_t\right) x_0 \right) \delta(x - a_t x_0 + b_t x_1) \rho_0(x_0) \rho_1(x_1) dx_0 dx_1 \\
&= \frac{\dot{b}_t}{b_t} x \rho_t(x) + \left(\dot{a}_t - \frac{\dot{b}_t}{b_t} a_t\right) \int_{\mathbb{R}^d \times \mathbb{R}^d} x_0 \delta(x - a_t x_0 + b_t x_1) \rho_0(x_0) \rho_1(x_1) dx_0 dx_1
\end{aligned}
$$
$$\text{(F.1)}$$

If $\rho_0(x_0) = (2\pi)^{-d/2} e^{-\frac{1}{2}|x_0|^2}$, we have the identity $x_0 \rho_0(x_0) = -\nabla_{x_0} \rho_0(x_0)$. Inserting this equality in the last integral in (F.1) and integrating by part using

$$\nabla_{x_0} \delta(x - a_t x_0 + b_t x_1) = -a_t \nabla_x \delta(x - a_t x_0 + b_t x_1) \quad \text{(F.2)}$$

gives

$$j_t(x) = \frac{\dot{b}_t}{b_t} x \rho_t(x) - a_t \left(\dot{a}_t - \frac{\dot{b}_t}{b_t} a_t\right) \nabla \rho_t(x) \quad \text{(F.3)}$$

This means that

$$v_t(x) = \frac{\dot{b}_t}{b_t} x - a_t \left(\dot{a}_t - \frac{\dot{b}_t}{b_t} a_t\right) \nabla \log \rho_t(x) \quad \text{(F.4)}$$

Solving this expression in $\nabla \log \rho_t(x)$ and specializing it to the trigonometric interpolant with $a_t, b_t$ given in (B.4) gives the first equation in (28). The second one can be obtained by taking the limit of this first equation using $v_{t=1}(x) = 0$ from (B.20) and l'Hôpital's rule.    □

Note that (F.3) shows that, when the interpolant is of the type (B.2) and $\rho_0(x_0) = (2\pi)^{-d/2} e^{-\frac{1}{2}|x_0|^2}$, the continuity equation (3) can also be written as the diffusion equation

$$\partial_t \rho_t(x) + \frac{\dot{b}_t}{b_t} \nabla \cdot (x \rho_t(x)) = a_t \left(\dot{a}_t - \frac{\dot{b}_t}{b_t} a_t\right) \Delta \rho_t(x) \quad \text{(F.5)}$$

Since we assume that $\dot{a}_t \leq 0$ and $\dot{b}_t \geq 0$ (see (B.3)), the diffusion coefficient in this equation is negative

$$a_t \left(\dot{a}_t - \frac{\dot{b}_t}{b_t} a_t\right) \leq 0 \quad \text{(F.6)}$$

This means that (F.5) is well-posed backward in time, i.e. it corresponds to backward diffusion from $\rho_{t=1} = \rho_1$ to $\rho_{t=0} = \rho_0 = (2\pi)^{-d/2} e^{-\frac{1}{2}|x_0|^2}$. Therefore, reversing this backward diffusion, similar to what is done in score-based diffusion models, gives an SDE that transforms samples from $\rho_0$ to $\rho_1$. Interestingly, these forward and backward diffusion processes arise on the finite time interval $t \in [0, 1]$; notice however that both the drift and the diffusion coefficient are singular at $t = 1$. This is unlike the velocity $v_t(x)$ which is finite at $t = 0, 1$ and is given by (B.19).

## G    GENERALIZED INTERPOLANTS

Our construction can be easily generalized in various ways, e.g. by making the interpolant depend on additional latent variables to be averaged upon. This enlarges the class of interpolant density we can construct, which may prove useful to get simpler (or more optimal) velocity fields in the continuity equation (3). Let us consider one specific generalization of this type:

**Factorized interpolants.** Suppose that we decompose both $\rho_0$ and $\rho_1$ as

$$\rho_0(x) = \sum_{k=1}^K p_k \rho_0^k(x), \qquad \rho_1(x) = \sum_{k=1}^K p_k \rho_1^k(x), \tag{G.1}$$

where $K \in \mathbb{N}$, $\rho_0^k$ and $\rho_1^k$ are normalized PDF for each $k = 1, \ldots, K$, and $p_k > 0$ with $\sum_{k=1}^K p_k = 1$. We can then introduce $K$ interpolants $I_t^k(x_0, x_1)$ with $k = 1, \ldots, K$, each satisfying (4), and define the stochastic interpolant

$$x_t = I_t^k(x_0, x_1), \qquad k \sim p_k, \quad x_0 \sim \rho_0^k, \quad x_1 \sim \rho_1^k \tag{G.2}$$

This corresponds to splitting the samples from $\rho_0$ and $\rho_1$ into $K$ (soft) clusters, and only interpolating between samples in cluster $k$ in $\rho_0$ and samples in cluster $k$ in $\rho_1$. This clustering can either be done beforehand, based on some prior information we may have about $\rho_0$ and $\rho_1$, or be learned (more on this point below).

It is easy to see that the PDF $\rho_t$ of $x_t$ is formally given by

$$\rho_t(x) = \sum_{k=1}^K p_k \int_{\mathbb{R}^d \times \mathbb{R}^d} \delta(x - I_t^k(x_0, x_1)) \rho_0^k(x_0) \rho_1^k(x_1) dx_0 dx_1, \tag{G.3}$$

and that this density satisfies the continuity equation (B.11) for the current

$$j_t(x) = \sum_{k=1}^K p_k \int_{\mathbb{R}^d \times \mathbb{R}^d} \partial_t I_t^k(x_0, x_1) \delta(x - I_t^k(x_0, x_1)) \rho_0^k(x_0) \rho_1^k(x_1) dx_0 dx_1, \tag{G.4}$$

Therefore this equation can be written as (3) with the velocity $v_t(x)$ which is the unique minimizer of a generalization of the objective (9). We state this as:

**Proposition G.1.** *The stochastic interpolant $x_t$ defined in (G.2) with $I_t^k(x_0, x_1)$ satisfying (4) for each $k = 1, \ldots, K$ has a probability density $\rho_t(x)$ that satisfies the continuity equation (3) with a velocity $v_t(x)$ which is the unique minimizer over $\hat{v}_t(x)$ of the objective*

$$G_K(\hat{v}) = \sum_{k=1}^K p_k \int_0^1 \int_{\mathbb{R}^d \times \mathbb{R}^d} \left( |\hat{v}_t(I_t^k)|^2 - 2\partial_t I_t^k \cdot \hat{v}_t(I_t^k)) \right) \rho_0^k(x_0) \rho_1^k(x_1) dx_0 dx_1 dt \tag{G.5}$$

*where we used the shorthand notations $I_t^k = I_t^k(x_0, x_1)$ and $\partial_t I_t^k = \partial_t I_t^k(x_0, x_1)$. In addition the minimum value of this objective is given by*

$$G_K(v) = -\int_0^1 \int_{\mathbb{R}^d} |v_t(x)|^2 \rho_t(x) dx dt > -\infty \tag{G.6}$$

*and both these statements remain true if $G(\hat{v})$ is minimized over velocities that are gradient fields, in which case the minimizer is of the form $v_t(x) = \nabla \phi_t(x)$ for some potential $\phi_t : \mathbb{R}^d \to \mathbb{R}$ uniquely defined up to a constant.*

We omit the proof of this proposition since it is similar to the one of Proposition 1. The advantage of this construction is that it gives us the option to make the transport more optimal by maximizing $G_K(v)$ over $I_t^k$ and/or the partitioning used to define $\rho_0^k$, $\rho_1^k$, and $p_k$. For example, if we know that the target density $\rho_1$ has $K$ clusters with relative mass $p_k$, we can define $\rho_0$ as a Gaussian mixture with $K$ modes, set the weight of mode $k$ to $p_k$, and maximize $G_K(v) = \min_{\hat{v}} G_K(\hat{v})$ over the mean $m_k \in \mathbb{R}^d$ and and the covariance $C_k \in \mathbb{R}^d \times \mathbb{R}^d$ of each mode $k = 1, \ldots, K$ in the mixture density $\rho_0$.

# H EXPERIMENTS FOR OPTIMAL TRANSPORT, PARAMETERIZING $I_t$, PARAMETERIZING $\rho_0$

As discussed in Section 2, the minimizer of the objective in equation (10) can be maximized with respect to the interpolant $I_t$ as a route toward optimal transport. We motivate this by choosing a parametric class for the interpolant $I_t$ and demonstrating that the max-min optimization in equation

(D.10) can give rise to velocity fields which are easier to learn, resulting in better likelihood estimation. The 2D checkerboard example is an appealing test case because the transport is nontrivial. In this case, we train the same flow as in Section 3.1, with and without optimizing the interpolant. We choose a simple parametric class for the interpolant given by a Fourier series expansion

$$\hat{I}_t(x_0, x_1) = \hat{a}_t x_0 + \hat{b}_t x_1 \tag{H.1}$$

where parameters $\{\alpha, \beta\}_{m=1}^M$ define learnable $\hat{a}_t, \hat{b}_t$ via

$$\hat{a}_t = \cos\left(\tfrac{1}{2}\pi t\right) + \frac{1}{M} \sum_{m=1}^M \alpha_m \sin(m\pi t), \qquad \hat{b}_t = \sin\left(\tfrac{1}{2}\pi t\right) + \frac{1}{M} \sum_{m=1}^M \beta_m \sin(m\pi t). \tag{H.2}$$

This is but one set of possible parameterizations. Another, for example, could be a rational quadratic spline, so that the endpoints for $\hat{a}_t$, $\hat{b}_t$ can be properly constrained as they are above in the Fourier expansion. In Figure H.1, we plot the log likelihood, the learned interpolants $\hat{a}_t$, $\hat{b}_t$ compared to their initializations, as well as the path length as it evolves over training epochs. With the learned interpolants, the path length is reduced, and the resultant velocity field under the same number of training epochs endows a model with better likelihood estimation, as shown in the left plot of the figure. For the path optimality experiments, $M = 7$ Fourier coefficients were used to parameterize the interpolant. This suggests that minimizing the transport cost can create models which are, practically, easier to learn.

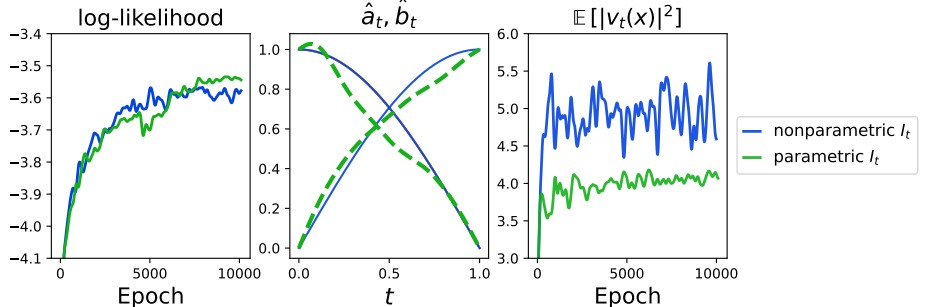

Figure H.1: Comparison of characteristics and performance of the learned vs nonparametric interpolant on the checkerboard density estimation. Left: Comparison of the evolution of the log-likelihood. Middle: The initial versus learned interpolant terms $\hat{a}_t, \hat{b}_t$. Right: The path length of the learned vs nonparametric interpolant.

In addition to parameterizing the interpolant, one can also parameterize the base density $\rho_0$ in some simple parametric class. We show that including this in the min-max optimization given in equation (D.10) can further reduce the transport cost and improve the final log likelihood. The results are given in Figure H.2. We train an interpolant just as described above, but also allow the base density $\rho_0$ to be parameterized as a Gaussian with mean $\hat{\mu}$ and covariance $\hat{\Sigma}$. The inclusion of the learnable base density results in a significantly reduced path length, thereby bringing the model closer to optimal transport.

## I  IMPLEMENTATION DETAILS FOR NUMERICAL EXPERIMENTS

Let $\{x_0^i\}_{i=1}^N$ be $N$ samples from the base density $\rho_0$, $\{x_1^j\}_{j=1}^n$ $n$ samples from the target density $\rho_1$, and $\{t_k\}_{k=1}^K$ $K$ samples from the uniform density on $[0, 1]$. Then an empirical estimate of the objective function in (9) is given by

$$G_{N,n,K}(\hat{v}) = \frac{1}{KnN} \sum_{k=1}^K \sum_{j=1}^n \sum_{i=1}^N \left| \hat{v}_{t_k}\left(I_{t_k}(x_0^i, x_1^j)\right) \right|^2 - 2\partial_t I_{t_k}(x_0^i, x_1^j) \cdot \hat{v}_{t_k}(I_{t_k}(x_0^i, x_1^j)). \tag{I.1}$$

This calculation is parallelizable.

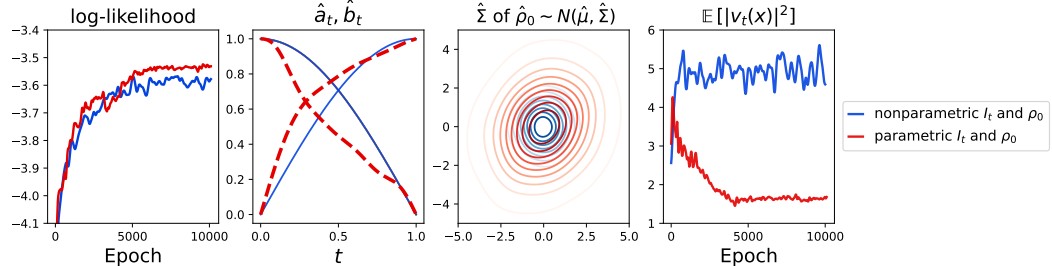

Figure H.2: Comparison of characteristics and performance of the learned vs nonparametric interpolant on the checkerboard density estimation, while also optimizing the base density $\hat{\rho}_0$. The parametric $\hat{\rho}_0$ is given as bivariate Gaussian $\mathcal{N}(\hat{\mu}, \hat{\Sigma})$. Left: Comparison of the evolution of the log-likelihood. Center left: The initial versus learned interpolant terms $\hat{a}_t, \hat{b}_t$. Center right: The learned covariance $\hat{\Sigma}$ in red compared to the original identity covariance matrix. Right: The path length of the learned vs nonparametric interpolant.

.

|  | POWER | GAS | HEPMASS | MINIBOONE | BSDS300 |
|---|---|---|---|---|---|
| Dimension | 6 | 8 | 21 | 43 | 63 |
| # Training point | $1,615,917$ | 852,174 | 315,123 | 29,556 | $1,000,000$ |
| Target batch size | 800 | 800 | 800 | 800 | 300 |
| Base batch size | 150 | 150 | 150 | 150 | 200 |
| Time batch size | 10 | 10 | 20 | 10 | 20 |
| Training Steps | $10^5$ | $10^5$ | $10^5$ | $10^5$ | $10^5$ |
| Learning Rate (LR) | 0.003 | 0.003 | 0.003 | 0.003 | 0.002 |
| LR decay (4k epochs) | 0.8 | 0.8 | 0.8 | 0.8 | 0.8 |
| Hidden layer width | 512 | 512 | 512 | 512 | 1024 |
| # Hidden layers | 4 | 5 | 4 | 4 | 5 |
| Inner activation functions | ReLU | ReLU | ReLU | ReLU | ELU |
| Beta $\alpha, \beta$, time samples | (1.0,0.5) | (1.0,0.5) | (1.0,0.5) | (1.0,1.0) | (1.0,0.7) |

Table 3: Hyperparameters and architecture for tabular datasets.

The architectural information and hyperparameters of the models for the resulting likelihoods in Table 2 is presented in Table 3. ReLU (Nair & Hinton, 2010) activations were used throughout, barring the BSDS300 dataset, where ELU (Clevert et al., 2016) was used. Table formatting based on (Durkan et al., 2019).

In addition, reweighting of the uniform sampling of time values in the empirical calculation of (I.1) was done using a Beta distribution under the heuristic that the flow should be well trained near the target. This is in line with the statements under Proposition 1 that any weight $w(t)$ maintains the same minimizer.

The details for the image datasets are provided in Table 4. We built our models based off of the U-Net implementation provided by lucidrains public diffusion code, which we are grateful for https://github.com/lucidrains/denoising-diffusion-pytorch. We use the sinusoidal time embedding, but otherwise use the default implementation other than changing the U-Net dimension multipliers, which are provided in the table. Like in the tabular case, we reweight the time sampling to be from a beta distribution. All models were implemented on a single A100 GPU.

## I.1 DETAILS ON COMPUTATIONAL EFFICIENCY AND DEMONSTRATION OF CONVERGENCE GUARANTEE

Below, we show that the results achieved in 3 are driven by a model that can train significantly more efficiently than the maximum likelihood approach to ODEs. Following that, we provide an illustration of the convergence requirements on the objective defined in (11).

|  | CIFAR-10 | ImageNet 32×32 | Flowers |
|---|---|---|---|
| Dimension | 32×32 | 32×32 | 128×128 |
| # Training point | $5 \times 10^4$ | 1,281,167 | 315,123 |
| Batch Size | 400 | 512 | 50 |
| Training Steps | $5 \times 10^5$ | $6 \times 10^5$ | $1.5 \times 10^5$ |
| Hidden dim | 256 | 256 | 256 |
| Learning Rate (LR) | 0.0001 | 0.0002 | 0.0002 |
| LR decay (1k epochs) | 0.985 | 0.985 | 0.985 |
| U-Net dim mult | [1,2,2,2,2] | [1,2,2,2] | [1,1,2,3,4] |
| Beta $\alpha, \beta$, time samples | (1.0,0.75) | (1.0,0.75) | (1.0,0.75) |
| Learned $t$ sinusoidal embedding | Yes | Yes | Yes |
| # GPUs | 1 | 1 | 1 |

Table 4: Hyperparameters and architecture for image datasets.

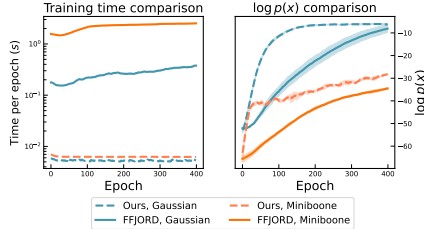

Figure I.1: *Left*: Training speed for ours vs. MLE ODE flows, with 400x speedup on the MiniBooNE. *Right*: InterFlow shows more efficient likelihood ascent.

We briefly describe the experimental setup for testing the computational efficiency of our model as compared to the FFJORD maximum likelihood learning method. We use the same network architectures for the interpolant flow and the FFJORD implementation, taking the architectures used in that paper. For the Gaussian case, this is a 3 layer neural network with hidden widths of 64 units; for the 43-dimensional MiniBooNE target, this is a 3 layer neural network with hidden widths of 860 units.

Figure I.1 shows a comparison of both the cost per training epoch and the convergence of the log likelihood across epochs. We take the architecture of the vector field as defined in the FFJORD paper for the 2-dimensional Gaussian and MiniBooNE, and use it to define the vector field for the interpolant flow. The left side of Figure I.1 shows that the cost per iteration is constant for the interpolant flow, while it grows for MLE based approaches as the ODE gets more complicated to solve. The speedup grows with dimension, 400x on MiniBooNE.

The right side of Figure I.1 shows that, under equivalent conditions, the interpolant flow can converge faster in number of training steps, in addition to being cheaper per step. The extent of this benefit is dependent on both hyperparameters and the dataset, so a general statement about convergence speeds is difficult to make. For the sake of this comparison, we averaged over 5 trials for each model and dataset, with variance shown shaded around the curves.

As described in Section 1, the minimum of $G(\hat{v})$ is bounded by the square of the path taken by the map $X_t(x)$. The shifted value of the objective $\tilde{G}(\hat{v})$ in (11) can be tracked to ensure that the model velocity $\hat{v}_t(x)$ meets the requirement of the objective. It must be the case that $\tilde{G}(\hat{v}) = 0$ if $\hat{v}_t(x)$ is taken to be the minimizer of $G(\hat{v})$, so we can look for this signature during the training of the

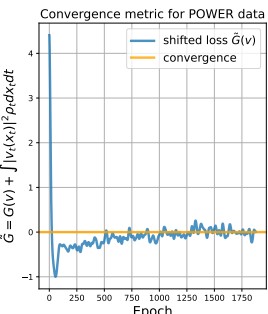

Figure I.2: Demonstration of the convergence diagnostic on POWER dataset. This is necessary but not sufficient for convergence. See Section 2 for definition of $\tilde{G}$.

interpolant flow. Figure I.2 displays this phenomenon for an interpolant flow trained on the POWER dataset. Here, the shifted loss converges to the target $\tilde{G}(v) = 0$ and remains there throughout training. This suggests that the dynamics of the stochastic optimization of $G(\hat{v})$ are dual to the squared path length of the map.

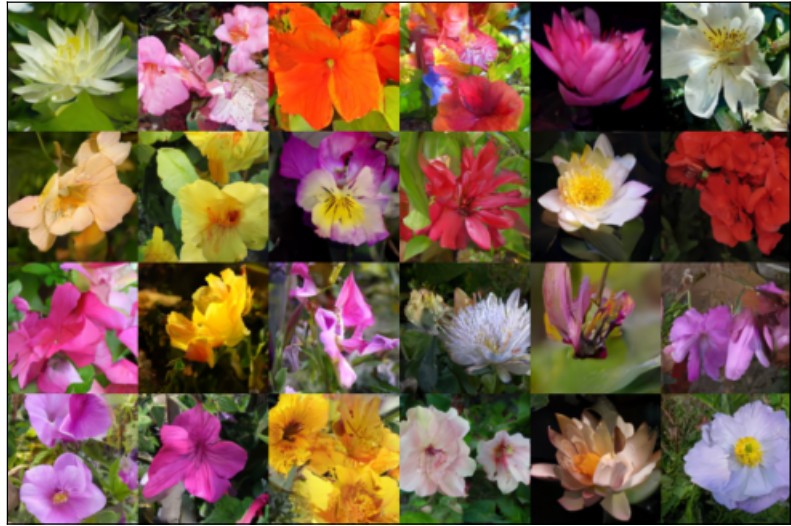

Figure I.3: Uncurated samples from Oxford Flowers 128x128 model.

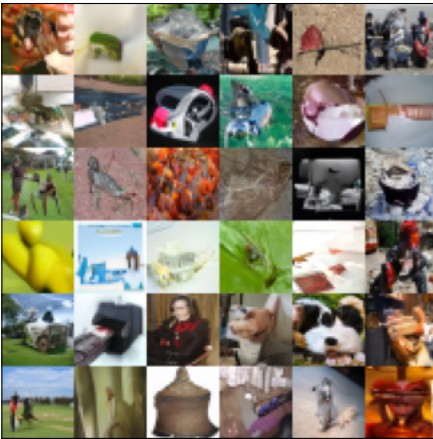

Figure I.4: Uncurated samples from ImageNet 32×32 model.

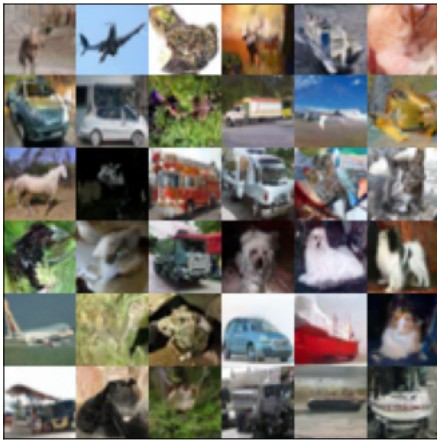

Figure I.5: Uncurated samples from CIFAR-10 model.

