# OpenReview forum: "Building Normalizing Flows with Stochastic Interpolants"
_ICLR.cc/2023/Conference — ICLR 2023 poster_

### Official Review · Reviewer_ogEs · 2022-10-21

**Confidence:** 3
**Correctness:** 4
**Technical Novelty And Significance:** 3
**Empirical Novelty And Significance:** 4
**Recommendation:** 8

**Clarity, Quality, Novelty And Reproducibility:**

All four aspects are excellent: clear, high-quality write-up, a novel idea, and plenty of experimental details throughout the main text.

**Strength And Weaknesses:**

Strengths:
- An idea that echoes prior work (pre-defining a sequence of densities and fitting it), yet is sufficiently novel and required a non-trivial amount of mathematical reasoning.
- Further theoretical study of the method, deriving a bound on the density fit, as well as showing a connection to the important class of generative models (diffusion).
- Experimental results: clear improvement on competing continuous flows on tabular data. Sensible results on synthetic datasets.
- High-quality, polished write-up with crisp notation.
- An in-depth discussion and analysis of the computation efficiency of the method.

Weaknesses/questions:
- Motivation for the interpolation function chosen in the paper: why have authors not started with a simpler, e.g. linear, interpolation function? Are there any preliminary experiments with alternative interpolation functions to discuss, and especially whether the method is sensitive to this choice?
- Diffusion-based models are known for their excellent sample quality. Given the shown connection between two classes of models, have authors tried fitting their model to higher-dimensional image datasets, and evaluating the samples?
- Fundamentally, the method aims to pre-define a continuous sequence of densities, and recover the dynamics from this sequence. Have authors considered alternative ways of constructing this sequence? Does the section 2.2 suggest that we could use a diffusion process instead?

**Summary Of The Paper:**

The paper proposes an alternative learning objective for a continuous normalizing flow. Instead of standard maximum likelihood training, which requires backpropagating through the ODE solver, authors define a continuous sequence of intermediate densities between $p_0$ and $p_1$, and train the dynamics of the continuous flow to match this sequence. The intermediate densities are defined using a differentiable interpolation function. The dynamics of the flow are trained to minimize an objective function that only requires samples from the base density, samples from the target density, and samples of time $t$. Authors show that this objective has a unique minimum, which is achieved when the _true_ dynamics are recovered. In addition, authors derive a bound on the Wasserstein distance between the learned density and the true target density, as well as link the proposed model to score-based diffusion models. Numerical experiments are undertaken which demonstrate the effectiveness of the method on real tabular data.

**Summary Of The Review:**

I strongly recommend accepting the paper. The proposed method draws inspiration from a set of methods in the field, yet is sufficiently novel, and is analyzed in-depth, both theoretically and empirically (albeit the empirical part could be even stronger, see suggestions above). Experimental results are strong, and the manuscript itself is excellent.

---

> ### Author Response · Authors · 2022-11-19
> **Response to Reviewer ogEs**
>
> We are glad you like the work and find the exposition of its unique points clear. We are happy to provide more details to your comments.
>
>
> **Motivation for the interpolation function chosen in the paper:**
>
> This is a great question, and the interpolant picture allows us to theoretically motivate it. In the case that the base and target are Gaussian mixtures, the exact velocity field can be derived. In Appendix C, we show that the transport in this case preserves the variance of the intermediate distribution, which is an appealing point of control.
>
> Moreover, the interpolant picture allows us to directly formulate *a procedure for optimal transport*, stated in Section 2 and described in Appendix D, by maximizing the the objective in eq ($9$) with respect to the interpolant. We provide a derivation of this and some simple experiments. We quite like this result.
>
>
> **Image experiments**
>
> You are right to wonder about image experiments. We have included some results for unconditional image generation, comparing with diffusion methods on CIFAR10 and ImageNet and even scaling ab-initio ODE solvers up to 128x128 images on the Oxford Flowers Dataset. All training was done on a single A100 GPU. Please see Section 3.4 for a discussion.
>
>
> **Connection to diffusions**
>
> As we mention in Sec. 2.2 and discuss in detail in Appendix F, provided that either the base or the target is a Gaussian density, we can then *a posteriori* turn this ODE into an SDE, but we don't have to (and may not want to since the drift and the diffusion in the resulting SDE are singular at one of the end point, unlike the velocity of the probability flow ODE). Note that in that setup, we can also recover the score, which may be useful for resampling using Eq. (2.6).

---

### Official Review · Reviewer_KfAo · 2022-10-23

**Confidence:** 4
**Correctness:** 4
**Technical Novelty And Significance:** 3
**Empirical Novelty And Significance:** 2
**Recommendation:** 5

**Clarity, Quality, Novelty And Reproducibility:**

The writing is clear and the main ideas are clearly exposed.

As emphasized in the previous comment, the approach is novel with the caveat that one important and very related work is not discussed.

Regarding the quality of the work. I believe the methodology and theory developed in the paper to be interesting. However the experiments are a weak spot of the paper and more experimental investigation is required.

No anonymous repository is shared but the authors provide enough details about the experiments so that I believe them to be reproducible

**Details Of Ethics Concerns:**

No ethical concern.

**Strength And Weaknesses:**

STRENGHTS:
* I think this is a nice idea that showcases how the manipulation of (Fokker-Planck) PDEs can yield new algorithms. I thought the paper was well written with the main contribution clearly presented. It extends denoising diffusion models in a new direction by focusing on the issue of the speed in these models.
* The theoretical results regarding the control in Wasserstein 2 are new.

WEAKNESSES:
* The experiments are quite weak. Considering that the presented approach extends diffusion models I would have expected the authors to compare their approach to diffusion models at some point. This impression is reinforced by the fact that "The choice of interpolant for experimentation was universally selected to be that of (5)" (where (5) highlights the link with diffusion models). It is not clear to me that the current approach brings any benefit compared to diffusion models. I think that the authors miss the opportunity to compare and evaluate different interpolants. Indeed, they introduce a whole new framework but end up only using one interpolant, which is underwhelming.

* I find a bit misleading that bold numbers correspond to only the best methods among continuous normalizing flows (CNF) in Table 1, since we can find that other normalizing flows yield better results for each considered task. I have read that the authors "primary point of comparison is to other continuous time models, so we sequester them in benchmarking" but no explanation is given as to why CNFs perform worse than NFs for tabular data and what could be done to remedy this situation. Also, confidence intervals are missing in Table 1.

* The authors do not discuss some related literature. In particular, one work that seems to be particularly relevant is [1] which considers a similar approach to the one of the author. Indeed in [1] the author considers a stochastic interpolant between $x_0$ and $x_T$ given by a bridge (in this paper a Stochastic Differential Equation (SDE)). Then the obtained diffusion is amortized with respect to some coupling between $\pi_0$ and $\pi_T$. The author of [1] shows that the obtained dynamics can also be described by an SDE whose drift can be learned. Finally, I think that the authors miss to discuss extension of denoising diffusion models which also consider finite-time dynamics like [2,3,4]. Is there any link between the proposed approach and control and optimal transport?

* Regarding the result in Wasserstein 2. Am I correct assuming that the exponential bound should depend on the finite time $T$? The current bound is correct because the authors restrict themselves to the $[0,1]$ interval but any rescaling of the time would harm the bound. This exponential dependency comes from the use of a Gronwall lemma and should be discussed.

OTHER COMMENTS/QUESTIONS:

* What limits the authors to consider an amortization w.r.t. to the independent coupling in Equation (6)? One could imagine other couplings and I think that the obtained results would still be true. Am I correct assuming that? In that case, a link with diffusion models can be drawn in a more explicit fashion by considering the coupling given by $X_0 \sim \pi_0$ and $X_T = e^-T X_0 + (1- e^{-2T})^{1/2} Z$ with $Z$ a Gaussian random variable. For $T= +\infty$ we recover the independent coupling but diffusion models are always run up to a finite time $T$.

* The current work is limited to ODEs. From a generative modeling point of view it has been shown [3] that ODEs and deterministic methods in general yield worse results than their SDEs counterpart. Can the theory developped by the authors be extended to the SDE setting? I am guessing that in that case the authors would recover derivations similar to [1].

[1] Peluchetti -- Non-Denoising Forward-Time Diffusions

[2] Vargas, Thodoroff, Lawrence, Lamacraft - Solving Schrödinger Bridges via Maximum Likelihood

[3] De Bortoli, Thornton, Heng, Doucet - Diffusion Schrödinger Bridge with Applications to Score-Based Generative Modeling

[4] Chen, Liu, Theodorou - Likelihood Training of Schrödinger Bridge using Forward-Backward SDEs Theory

[5] Karras, Aittala, Aila, Laine - Elucidating the Design Space of Diffusion-Based Generative Models

**Summary Of The Paper:**

In this paper, we authors propose a new way to interpolate between two probability distributions. This is done by constructing a stochastic interpolant between two points $x_0$ and $x_T$ distributed w.r.t. $\pi_0$ and $\pi_T$ respectively. It turns out that the amortized dynamics obtained by integration w.r.t. to the product measure $\pi_0 \pi_T$ satisfies a continuity equation and can also be related to an Ordinary Differential Equation (ODE). The drift of this ODE can be learned in a manner akin to the Implicit Score Matching (ISM) in diffusion models. The authors elaborate on the links with diffusion models by showing that for a specific choice of interpolants the estimated velocity is related to the score of a diffusion model. They evaluate the performance of their approach on several toy and tabular datasets.

**Summary Of The Review:**

To conclude, I think that the approach developed by the author is interesting.

The methodology and theory introduced in that paper are coherent and represent a nice extension of diffusion models.
However, I think that more investigation (in particular numerical) is required to make the case that this approach can be used 1) in a large-scale setting 2) outside of the diffusion models setting.

For these reasons I think that the work is currently not mature enough.

---

> ### Author Response · Authors · 2022-11-19
> **Response to Reviewer KfAo, Part 1**
>
> We thank you for your very thorough and thoughtful discussion of our work. We are glad that you like the approach. We have itemized your comments to clarify what we have added to the paper and how they address your remarks.
>
>
> **Improvements to experiments**
>
> We agree that a comparison to diffusion models is valuable, and we have supplied that in image generation experiments now given in Section 3.4. We benchmark the method against recent score-based diffusion papers on CIFAR-10 and ImageNet32, as is done in the relevant Variational Diffusion Models and the ScoreFlow papers. We benchmark on FID and negative log likelihood, and show competitive performance with these recent models. We supply some exemplary images from the model. We also include examples on the 128x128 Oxford flower dataset to show that we can generate realistic images at a large scale. Experiments were done on a single A100 GPU and **show the scalability of our approach**.
>
>
> **A clarification**
>
> We would like to clarify, however, that our approach is orthogonal to diffusion models and not an extension of them. To that end, we have included a table in the related work to lay out the distinguishing characteristics amongst these methods. Our aims were to work directly with the probability flow picture rooted in normalizing flows to show that there is a clean and general “simulation-free” path to training flows directly, instead of needing to rely on the SDE. We believe that there are benefits to what the theory behind this method brings, also summarized in the table in related work: finite time integration, learning a deterministic map in a simple quadratic loss, and computable likelihoods. It just so happens that when the base distribution is a Gaussian, the corresponding velocity field yields access to the score, through which diffusion based methods could be applied.
>
>
> **Motivating the trigonometric interpolant**
>
> The interpolant picture can provide some nice theoretical insight. For example, we can motivate the choice of the trigonometric interpolant in the following way, separate from the question of diffusions: In the appendix, we include derivations of the interpolant for Gaussian mixtures. In this case, we can directly write down the associated velocity field arising from the interpolant, which, for the trigonometric interpolant (5), yields a variance preserving map, i.e. the intermediate density $\rho_t$ has its variance preserved. This was the impetus for this choice of interpolant. With regards to considering other interpolants, we think this is a valuable point and address this below when discussing optimal transport with interpolant flows.
>
>
> **Removing bolding in results**
>
> Bolding was following the convention of Grathwohl et al 2019 (FFJORD) for which the direct comparison was originally being made. We agree that it is superfluous, as all the numbers are generally proximal to each other.
>
>
>
> **Related work on finite time diffusion bridges**
>
> We hope to provide some clarity on the relation of this work, thank you to pointing us to it, as we did not see a preprint or publication of it. We have included it in a diffusion bridges section of the text. However, we’d like to clarify a potential point of misconception (see also the reply to Reviewer xdhs). The stochastic part in the name of our stochastic interpolant is about the pairing of $x_0 \sim \rho_0$ with $x_1 \sim \rho_1$, not with the transport map that takes a sample from $\rho_0$ to a sample under $\rho_1$.
>
> The bridge picture of $[1]$ is a nice way to do this if one insists on using an SDE, but our apparoch shows that we do not have to: notice in particular that our stochastic interpolant is time-differentiable by construction, unlike the solution of an SDE or a diffusion bridge, and it leads to velocity fields that are nonsingular at the end points despite allowing for the *exact* connection beween any base and target densities.
>
> Since their aim is similar to bridge diffusions, we also now cite works on Schroedinger bridges and included a row about them in the table to distinguish it. Do let us know if this helps clarify things, as we are happy to try to address any remaining question.

---

> > ### Author Response · Authors · 2022-11-19
> > **Response to Reviewer KfAo, Part 2**
> >
> > **Is there any link between the proposed approach and control and optimal transport?**
> >
> >
> > There is indeed a link to optimal transport. We were considering that to be its own separate paper, but you have asked a good question and we have included a discussion about it in the present work. We have added a proof that by maximizing the objective over the interpolant, our procedure maps onto the Benamou-Brenier formulation of optimal transport. We have provided a statement of that in the main text, as well as derivations and demonstrations of the potential benefit of doing so in the appendix. This relates to a valuable point you made above about considering other interpolants. Directly parameterizing the interpolant, under the right conditions, is what can lead to optimal transport. We discuss this class of interpolants in the appendix.
> >
> >
> >
> > **Questions regarding the Wasserstein-2 bound, Gronwall Lemma**
> >
> > The Wasserstein bound does indeed involve Gronwall lemma, and the final time does not appear since we can fix it to be $t=1$ by construction (again, without this affecting the ability of our approach to connect the base to the target density exactly). Of course, the catch is in the Lipschtiz constant of the velocity field that we do not know, and could presumably be very big. This issue is however common to most bounds of the sort derived in the literature, and ultimately the effectivness of the approach must be tested on challenging examples (as we do now). Interstingly, the bound suggests that a way to regularize the objective, as specified in Eq.(E.7)
> >
> >
> > **Other couplings besides independent couplings**
> >
> > This is an insightful comment pointing out to an interesting way to look at the difference between SBDM and what we do. Using the coupling  $X_t=e^{−t}X_0+(1−e^{−2t})^{1/2}Z$ with $X_0∼π_0$ and $Z$ an *independent* Gaussian random variable is indeed a form of stochastic interpolant, albeit one that requires to take $t\to\infty$ to reach the Gaussian and is not differentiable at $t=0$ since setting $X_t = I_t(X_0,Z)$ we get $\partial_t I_t(X_0,Z) = - e^{-t} X_0 + (1−e^{−2t})^{-1/2} e^{-2t}Z$. Since this specific interpolant can be viewed as a solution to an SDE (it has the same law at any time $t$, albeit not the same paths), we can obtain the probability flow ODE via learning the score using SBDM.  Our work shows that we can avoid this altogether by modifying the interplant to (i) make it differentiable and (ii) guaranteeing that it connects the base and the target exactly in finite time, then learning the velocity field in the probability flow ODE directly. As we mention in Sec. 2.2 and discuss in detail in Appendix F, provided that either the base or the target is a Gaussian density, we can then *a posteriori* turn this ODE into an SDE, but we do not have to (and may not want to since the drift and the diffusion in the resulting SDE are singular at one of the end point, unlike the velocity). Note that in that setup, we can also recover the score, which may be useful for resampling using Eq. (2.6).
> >
> >
> > [1] Peluchetti -- Non-Denoising Forward-Time Diffusions

---

### Official Review · Reviewer_tuy2 · 2022-10-26

**Confidence:** 5
**Correctness:** 4
**Technical Novelty And Significance:** 4
**Empirical Novelty And Significance:** 2
**Recommendation:** 6

**Clarity, Quality, Novelty And Reproducibility:**

**Clarity**
The paper is clear and well written. The method is cleanly motivated and described.

**Originality**
The method as proposed seems novel to me, and I'm not aware of previous work. In fact, the approach neatly takes lessons from efficient training of flows through the diffusion model formulation and circles back to make possible the training of flows directly and efficiently without the notion of a diffusion process. The connection to the Wasserstein-2 distance and the ability to build a flow between two arbitrary distribution for which we have samples is also novel.

**Strength And Weaknesses:**

**Strengths**
The fundamental idea proposed in the paper is excellent. Diffusion models showed how to design normalizing flows which bridge the data and base distribution by definition, and how to train them efficiently without taking gradients through ODE or SDE dynamics. This paper takes that idea one step further and shows that such a flow can be designed without resorting to a diffusion process, and that training amounts to minimizing a simple quadratic loss analogously to how denoising score-matching facilitates score-matching without access to the ground-truth score. Moreover, the paper relates the quadratic objective to the Wasserstein-2 distance between the target distribution and the distribution defined by transporting the base under the model, and the authors also demonstrate that the proposed flows can bridge between any two distributions from which we have samples, not just a target distribution and tractable base distribution such as a Gaussian. These are all strong contributions.

I'd also like to highlight a key point made by the paper, namely 'By specifying an interpolant density, the method therefore separates the tasks of minimizing the objective from discovering a path between the base and target densities. This is in contrast with conventional maximum likelihood (MLE) training of flows where one is forced to couple the choice of path in the space of measures to maximizing the objective.' I think this is a significantly underappreciated point in the modern generative modeling literature, and explains many of the issues with vanilla flows and VAEs which have been to some extent addressed by diffusion models. I am glad the paper explicitly notes the importance of this decoupling.

**Weaknesses**
Unfortunately, the experimental evaluation is lacking.

The 2D experiments are useful as a sanity check, but are not convincing enough to warrant their own section. Moreover, the fact that the proposed method can formulate a flow between two arbitrary but is again only tested on 2D data is unfortunate. For example, a compelling experiment which immediately jumped to mind is image upsampling. Diffusion-based image upsamplers transport a Gaussian to the high resolution image conditioned on a naively upsampled image using e.g. bilinear interpolation. In the case of the proposed method, a flow could be formulated to bridge between the naively upsampled image and the ground-truth directly. An experiment like this would be a strong demonstration of the advantage of the interpolation approach.

In terms of the tabular data experiments, I appreciate that this benchmark of 5 datasets has been standard in the literature, but it was originally introduced in 2017 and is now quite outdated. I would question its usefulness beyond a sanity check for the method, similar to the toy experiments. For example, BSDS is a dataset of 8x8 black-and-white image patches whose train split contains 1 million examples. Diffusion models have been fit to datasets consisting of hundreds of millions or even billions of images at higher resolution. While it's not possible for all new methods to be tested at this scale, it's still necessary to compare on a reasonable-scale problem, ImageNet. Alternatively, design a new benchmark for tabular data. I appreciate this is a lot to ask, but at some point benchmarks become outdated and are no longer useful to track progress in modeling.

Finally, and related to the previous points, the comparison to FFJORD is worthwhile as a baseline, but surely the primary mode of comparison should be flows trained in the diffusion model setting? These are much more competitive, and can even be trained by maximum likelihood without the need to backprop through ODE dynamics. This would provide a much stronger baseline, and a more convincing argument that the proposed method provides an advantage.



**Summary Of The Paper:**

This paper proposes a continuous-time normalizing flow which does not need (i) to propagate gradients through ODE dynamics for training or (ii) to formulate the flow using a diffusion. The method is tested empirically on 2D data and standard tabular data benchmarks.

**Summary Of The Review:**

I'm torn between the compelling idea presented in the paper, and the lacking empirical evaluation. With more thorough experimental evaluation, this would be an excellent submission. As it stands, I would still lean towards acceptance because I think the idea deserves to be communicated.

---

> ### Author Response · Authors · 2022-11-19
> **Response to Reviewer tuy2**
>
> We thank you for your insights regarding our work, and appreciate your feedback. We have made the following revisions to the work to address your comments, itemized for clarity:
>
> **The inclusion of additional experiments and benchmarking compared to diffusion models**
>
> We agree that more experiments are beneficial and we have endeavored to include that. In the experimental section we have included image generation results: following the recent flow literature and time permitting, we trained interpolant flows on CIFAR-10 and ImageNet 32x32, as well as a 128x128 Oxford flower dataset to **demonstrate that our method can scale**. All of this training was done on a single A100 GPU. We benchmark their performance on negative log likelihood (NLL) and Frechet Inception Distance (FID), as is done in the recent ScoreFlow paper and the Variational Diffusion Models paper, methods which relate to our work by trying to connect to the ODE picture of the map.
>
> In Section 3.4, you can find these results and example images produced by the model. Experimental details are provided in the appendix.
>
>
> **Other revisions to the text to add clarity, context, and completeness:**
>
> Another reviewer asked how optimal transport can come into the picture here, and we have added a proof that by maximizing the objective $(9)$ over the interpolant, our procedure maps onto the Benamou-Brenier formulation of optimal transport. We have provided a statement of that in the main text, as well as derivations and demonstrations in simple settings of the potential benefit of doing so in Appendix D (see especially proposition D.2).
>
> We have included a table in the related work to consolidate the distinguishing characteristics of myriad methods so as to avoid confusion and help guide the related work discussion. In that regard, we have also added some citations on some previous work on simulation-free generative models that are limited to manifolds, low-dimensions, or have biased gradients. This helps clarify our work as a general framework for connecting two densities, which is scalable and non-domain specific.
>
> We have included theoretical results on Gaussian Mixtures in the appendix that motivate the use of the trigonometric interpolant, namely that it is variance preserving in the intermediate densities between $\rho_0$ and $\rho_1$ in this case.
>
>
>
> Do let us know if there is anything else we can address.

---

### Official Review · Reviewer_xdhs · 2022-11-04

**Confidence:** 4
**Correctness:** 3
**Technical Novelty And Significance:** 2
**Empirical Novelty And Significance:** 2
**Recommendation:** 3

**Clarity, Quality, Novelty And Reproducibility:**

* Clear writing. Easy to follow.
* Novely is limited, as detailed above.
* Experiments can be improved. For 2D density estimation, need to have additional baselines (such as ScoreFlow and FFJORD). For tabular data and computational efficiency, need to compare wit ScoreFlow and other works mentioned above. Current experiments cannot show clear advantage over existing simulation-free approaches.

**Strength And Weaknesses:**

## Strength

1. The proposed formulation is very clear and easy to understand. The proof is easy to follow.
2. The results of Wasserstein bounds offer valuable insights.

## Weaknesses

I have major concerns regarding novelty and the discussion of relevant work in this paper:

1. The motivation of training continuous normalizing flows without requiring numerical ODE simulation is not new. The same challenge has been tackled in many previous works, such as [1][2][3]. In particular, the closest work to this paper is perhaps ScoreFlow [3], which leverages score-based diffusion models to design a simulation-free approach for maximum likelihood training of continuous normalizing flows. This paper needs to include a comprehensive discussion on what sets this work apart from existing ones. For example, is there anything that was impossible before now becomes tangible with the new method? Does the performance improve in comparison to other simulation-free approaches? Additional comparisons with prior methods, such as ScoreFlow in [3], need to be included in the experimental section (Figure 2, Table 1,2 and Figure 4).

2. The proposed stochastic interpolant method is not entirely new. It can actually be considered as a special case of the result in [4], where SDEs are replaced by ODEs (by setting the diffusion coefficient to zero). Specifically, the interpolation function $I_t(x_0, x_1)$ in this work defines an ODE connecting $x_0$ and $x_1$, with the form $x'_t = \partial_t I_t(x_0, x_1)$. By averaging this ODE over the joint distribution of $(x_0, x_1)$, we can obtain another ODE that smoothly converts $\rho_0$ to $\rho_1$. This ODE can be derived with the techniques in [4], given by $x'_t = E[\partial_t I_t(x_0, x_1) | I_t(x_0, x_1)]$, which is equivalent to the velocity field $v_t( I_t(x_0, x_t))$, the target to be learned in this work.

3. The proposed stochastic interpolant approach was also implied by the work [5] (see Appendix D). This paper needs to clarify the connection.

## References

[1] Rozen, Noam, et al. "Moser flow: Divergence-based generative modeling on manifolds." Advances in Neural Information Processing Systems 34 (2021): 17669-17680.

[2] Ben-Hamu, Heli, et al. "Matching normalizing flows and probability paths on manifolds." arXiv preprint arXiv:2207.04711 (2022).

[3] Song, Yang, et al. "Maximum likelihood training of score-based diffusion models." Advances in Neural Information Processing Systems 34 (2021): 1415-1428.

[4] Peluchetti, Stefano. "Non-Denoising Forward-Time Diffusions." (2021).

[5] Salimans, Tim, and Jonathan Ho. "Progressive distillation for fast sampling of diffusion models." arXiv preprint arXiv:2202.00512 (2022).


**Summary Of The Paper:**

This work proposes to train continuous normalizing flows by specifying a continuum of distributions using an interpolation function. The neural network model is trained to estimate the expected velocity of the interpolation function though MSE minimization. This training approach does not require numerical simulation of ODEs, and is therefore much more efficient compared to conventional maximum likelihood approaches. The method can be viewed as a generalization of score-based diffusion models to non-Gaussian noise. Experiments on toy datasets and tabular datasets confirm that the resulting continuous normalizing flows obtain competitive performance.

**Summary Of The Review:**

Overall nice discussion of a clever idea for training continuous normalizing flows (generative neural ODEs) without ODE simulation. However, the method is not entirely novel since it is a special case of an existing paper. Experiments are not comprehensive and more comparisons are needed to understand the trade-offs with existing simulation-free approaches.

---

> ### Author Response · Authors · 2022-11-18
> **Response to Reviewer xdhs**
>
>
> We thank you for pointing out several important references that were missing in the original submission. We agree that a more thorough discussion of these works, as is now given in the revised version of our paper, helps contextualizing our approach and explaining its advantages. However, we believe that the statement that our method is *a special case of an existing paper* [Peluchetti's "Non-Denoising Forward-Time Diffusions" paper] is too strong, and we hope to clarify.
>
>
> **Clarification with respect to existing works and novelty of our approach**
> - The paper "Maximum likelihood training of score-based diffusion models" that introduces ScoreFlow is indeed simulation-free and it shows how to derive a continuous-time normalizing flow in the form of a probability flow ODE. However, the paper uses SBDM, i.e. it does not adress the issues of connecting arbitrary densities in finite time. Our construction shows how to do so.
> - The paper "Progressive distillation for fast sampling of diffusion models" introduces a clever construction to amortize the number of steps needed in the training and simulation of the SDE used in generative models, but these models still involve SDE that fit the general SBDM framework. Our construction is simpler overall: the simulation of the probability flow ODE we derive could potentially be distilled using similar ideas, but our numerical results show that it is not necessary to do so since we only need to integrate the ODE on a finite time interval $t\in[0,1]$. This is now demonstrated in the revised version of the paper on more complicated numerical tests, with a comparison with other simulation-free methods (more on this below)
>
> The  other three papers quoted by the reviewers do propose to connect more general densities in finite time, thereby addressing an issue with SBDM and generalizing its scope. However:
>
> - The papers "Moser flow: Divergence-based generative modeling on manifolds." and "Matching normalizing flows and probability paths on manifolds" use the Dacorogna-Moser construction to build a density interpolating between a base and a target. While this construction is important theoretically, and it can be used in low dimension, it is not scalable to higher dimension. It also gives probability flow ODE with velocity fields that are very singular in general (in particular if the supports of these densities are very different, which is the generic case in high dimension). Our interpolant construction does not suffer from these issue -- in particular, by construction, the path of the interplant density has finte length in W2 distance that is bounded by the quantity in Eq. (7) that we can access and control (see Lemma B.2).
> - The paper "Non-Denoising Forward-Time Diffusions" uses Brownian bridges. This is an appealing and quite general idea, but it is also diffusion-based and different from our approach, despite the claim made in the review. Indeed diffusion bridges are conditioned SDEs, with drift and diffusion coefficients constructed via Doob $h$-transform, that must be singular at one end point, and involve SDE path that are continuous but not differentiable. In contrast, our construction involves an interpolant that is time-differentiable everywhere by definition and leads to a velocity field that is non-singular at the end point.
>
>
> We have modified the discussion in the paper to clarify these points and cited all of these works. Do let us know if these explanations and modifications to the paper address your concern.
>
> **The inclusion of additional numerical examples ...**
>
> We agree that more experiments are beneficial and we have endeavored to include that. In the experimental section we have included image generation results: following the recent flow literature and time permitting, we trained interpolant flows on CIFAR-10 and ImageNet 32x32, as well as a 128x128 Oxford flower dataset to **demonstrate that our method can scale**. All of this training was done on a single A100 GPU. We benchmark their performance on negative log likelihood (NLL) and Frechet Inception Distance (FID), as is done in the recent ScoreFlow paper and the Variational Diffusion Models paper, methods which relate to our work by trying to connect to the ODE picture of the map.
>
> In Section 3.4, you can find these results and example images produced by the model. Experimental details are provided in the appendix.
>
> This last statement is also given to reviewer tuy2.

---

### Author Response · Authors · 2022-11-18
**Overview and Hello to the Reviewers**

We thank the reviewers for their careful reading of our manuscript and their constructive comments and suggestions to improve it! Our response is structured as follows. First, we give a general statement to contextualize the method based on your useful feedback. Then, we detail the improvements we have made to address individual reviewer comments. This is provided in the **Common Reply**. Following that, we will give a more detailed reply to each reviewer based on their specific feedback. All references to text sections and equations are with respect to the revised document. Any increase in rating would be greatly valued if you feel we have addressed your apprehensions.

---

### Author Response · Authors · 2022-11-18
**Common Reply**

As highlighted by reviewer tuy2, the main advantage of our interpolant method is that *it separates the tasks of minimizing the objective from discovering a path between the base and target densities*. This feature is also at the core of score-based diffusion models (SBDM), and it simplifies the learning procedure. The main difference between our approach and SBDM is that it provides us with:
- a simpler and more flexible way to build a path connecting any base and target densities in finite time (and with finite W2 length), that avoids completely the need to use a diffusion (SDE);
- a quadratic objective for the velocity of the probability flow ODE, that involves no spatial or temporal derivatives of this velocity and is readily amenable to empirical estimation.
- the first results of scaling an *ab-initio* ODE sampler to image resolutions in the 100s, even on a single A100.


While other works exist that generalize SBDM and allow the definition of such finite-time connection between arbitrary base and target densities (including Schroedinger bridges, non-denoising forward-time diffusion, and Dacorogna-Moser flows: these works are discussed in more detail in the individual replies below), we believe that none have the combined simplicity and flexibility of our interpolant construction. We provide specifics in the bullet points below.

In the revised version, we have clarified this point and put our results in perspective of the works just mentioned, highlighting the key differences between these approaches and ours. This is included in a simple table in the related works, as well as an expansion of the discussion in this section to expound on the citations the reviewers mentioned. Indeed, the inclusion of these references helps us clarify what is novel about our contribution.

**Relatedly, we have also included a demonstration of the scalability and flexibility of our approach through image generation experiments**, modeled after the tests done in ScoreFlow and related works like Variational Diffusion Models.
- We benchmark our method against these related techniques on CIFAR-10 and ImageNet 32x32 using the U-Net architectures from the Denoising Diffusion Probabilistic Models paper and show that they are competitive in terms of log-likelihood even without optimizing this quantity directly. Our FIDs are slightly behind the state of the art but are proximal, suggesting that finetuning our first pass on these datasets could push these models further.
- Moreover, we demonstrate that these models can scale to *128x128* image generation tasks even on a single GPU, shown on the Oxford flowers dataset for demonstration.

**On the theory side**, we have included the following additional results to make the theory more complete and based on your suggestions:

- *To instantiate the benefit of using nonlinear interpolant, we now show that the objective in Eq (9) can be adapted to approach the problem of optimal transport. Namely, maximizing the objective with respect to the interpolant yields a solution to the optimal transport problem in the Benamou-Brenier framework. This is now mentioned in Section 2 and discussed in Appendix D*. We had originally decided to not include this result in our submission, as the issue of optimal transport is somewhat tangential to the problem we address in the paper, but we agree that mentioning it here helps justify why we want the freedom of using more general interpolants.
- *The simplicity of our interpolant construction also gives us access to other theoretical insight. We derive the exact velocity field in the case of Gaussian Mixtures to motivate the choice of the trigonometric interpolant. See Appendix C.*
- *A simple generalization of the interpolant construction involving latent variables is now included in Appendix G*
- *Elaborating on the new results in Appendix D, Appendix H contains further experimental results testing how to optimize the transport by shortening the W2 path length from two perspectives: by optimizing either the interpolant or adjustable parameters in the  base density assuming we have some leeway to do so.*

---

### Decision · Program_Chairs · 2023-01-20

**Decision:**

Accept: poster

**Justification For Why Not Higher Score:**

Most reviewers find the approach of this paper interesting. The additional results from the authors demonstrate the results can scale. However, as multiple reviewers pointed out that the approach is not intrinsically novel (e.g., multiple existing methods also presented simulation-free approaches), the AC believes this justifies a poster presentation.

**Justification For Why Not Lower Score:**

Including the new results and added discussions, the AC thinks the paper has become stronger and worthy of publication.

**Metareview: Summary, Strengths And Weaknesses:**

This paper presents an alternative learning objective for a continuous normalizing flow between any pair of base and target distributions. The core idea is to train a neural network to estimate the expected velocity of the interpolation function with an MSE loss, thereby bypassing the costly numerical simulation of ODEs used in maximum likelihood training approaches.

Strength:
- Multiple reviewers praised the well-motivated method and clear polished writing.
- The results of Wasserstein bounds offer valuable insights.

Weakness:
- As Reviewer xdhs pointed out, the challenges of simulation-free training of continuous normalizing have been tackled in several prior works. The discussions of the proposed method in the context of these previous method needs improvement.
- Comparison with simulation-free approaches is missing.

The authors provided a detailed response to these concerns. Unfortunately, the reviewers do not engage with the discussions. The AC took a close look at the responses and believes that many of the concerns have been addressed or at least alleviated (e.g., discussions with respect to relevant prior work for clarifying the contribution of this work, additional comparison results). The AC thinks the paper has sufficient merits and recommends to accept.


**Note From Pc:**

if the above contains the word "oral" or "spotlight" please see: "oral" presentation means -> notable-top-5% and "spotlight" means -> notable-top-25%. As stated in our emails, we are disassociating presentation type from AC recommendations